# Finite-hillslope analysis of landslides triggered by excess pore water pressure: the roles of atmospheric pressure and rainfall infiltration during typhoons

Lucas Pelascini[1], Philippe Steer[1], Maxime Mouyen[2] and Laurent Longuevergne[1]

[1]Univ Rennes, CNRS, Géosciences Rennes - UMR 6118, 35000, Rennes, France
[2]Department of Space, Earth and Environment, Chalmers University of Technology, *SE-412 96 Gothenburg, Sweden*

*Correspondence to*: Lucas Pelascini (lucas.pelascini@univ-rennes1.fr)

**Abstract.**

Landslides are often triggered by catastrophic events, among which earthquakes and rainfall are the most depicted. However,
very few studies have focused on the effect of atmospheric pressure on slope stability, even though weather events such as
typhoons are associated with significant atmospheric pressure changes. Indeed, both atmospheric pressure changes and rainfall-
induced groundwater level change can generate large pore pressure changes.   In this paper, we assess the respective impacts
of atmospheric effects and rainfall over the stability of a hillslope. An analytical model of transient groundwater dynamics is
developed to compute slope stability for finite hillslopes. Slope stability is evaluated through a safety factor based on the Mohr-
Coulomb failure criterion. Both rainfall infiltration and atmospheric pressure variations, which impact slope stability by
modifying the pore pressure of the media, are described by diffusion equations. The models have then been forced by weather
data from different typhoons that were recorded over Taiwan. While rainfall infiltration can induce pore pressure change up
to hundreds of kPa, its effects are delayed in time due to flow and diffusion. To the contrary, atmospheric pressure change
induces pore pressure changes not exceeding a few kPa, which propagates instantaneously through the skeleton before
diffusion leads to an effective decay of pore pressure. Moreover, the effect of rainfall infiltration on slope stability decreases
towards the toe of the hillslope and is cancelled where the water table reaches the surface, leaving atmospheric pressure change
as the main driver of slope instability. This study allows for a better insight of slope stability through pore pressure analysis,
and shows that atmospheric effects shouldn't always be neglected.

## 1 Introduction

In mountainous areas, landslides represent a major erosional process that contribute to landscape dynamics and frequently
cause significant damage and losses when catastrophic failures occur (Keefer, 1994; Malamud et al., 2004). Landslides can be
triggered by dynamic events, including earthquakes and storms, which drive hillslopes towards instability and catastrophic
failure (Haneberg, 1991; Iverson, 2000; Collins and Znidarcic, 2004; Hack et al., 2007). These two types of triggering events
have been extensively studied with numerous observations, empirical, analogical, numerical, and theoretical models.

Triggering of co-seismic (i.e., during an earthquake) landslides is generally attributed to the peak ground acceleration generated by seismic waves, but more complex phenomena comes into play, such as a cohesion loss, liquefaction, or topographic site effect (Hack et al., 2007; Meunier et al., 2007, 2008). Triggering of landslides by weather events involves various processes that are generally linked to rock-water interactions. Characterising and understanding how weather events trigger devastating landslides is essential (Baum et al., 2010; Rossi et al., 2012; Chen et al., 2014; Martha et al., 2015). At long time scales, weathering process affect rock mechanical properties, through chemical alteration. This rock-weakening process is known to reduce the slope stability and increase the risk of landslides (Calcaterra and Parise, 2010; Hencher and Lee, 2010). At monthly to seasonal time scales, groundwater recharge increases water table height and the pore pressure, which alters slope stability. As the wet season increases the groundwater level, this results in seasonal increase in frequency of catastrophic landslides – namely sudden failures leading to significant mass displacement (Gabet et al., 2004). At shorter time scales, water infiltration leads to a pressure front that modifies pore pressure and diffuses through the hillslope subsurface leading to its destabilization (Haneberg, 1991; Iverson, 2000; Collins and Znidarcic, 2004; Tsai and Yang, 2006). Large infiltration rates and high groundwater flow gradients can also generate seepage forces that further destabilise the slope (Budhu and Gobin, 1996).

Weather events are also characterised by a drop in atmospheric pressure which could influence slope stability. This slope destabilisation factor has received little attention. Indeed, atmospheric pressure changes induce a pressure differential at the water table, which results into pore pressure evolution via diffusion in the saturated zone until equilibrium with atmospheric pressure, modifying slope stability (Schulz et al., 2009). A correlation has been observed between atmospheric tides, leading to diurnal and semidiurnal atmospheric pressure changes, and displacement rate in a slow-moving landslide (Schulz et al., 2009). The amplitude of these repetitive pressure changes induced by atmospheric tides greatly depends on the latitude, but does not exceed 1.3 hPa around the equator (Lindzen and Chapman, 1969; Dai and Wang, 1999). Other atmospheric events can lead to much larger changes in atmospheric pressure. Indeed, typhoons and major storms can yield atmospheric drop of tens of hPa, which could in turn significantly alter the stability of slopes.

In this context, groundwater plays a crucial role in converting both atmospheric and rainfall-induced effects into mechanical pressure changes. Most of the studies using analytical models to represent slope stability use a 1D infinite slope model (Collins and Znidarcic, 2004; Iverson, 2000). However, modelling the full hillslope allows for a better characterisation of the evolution of groundwater level along the hillslope through modelling of the lateral flow. Since landslides are not evenly distributed along hillslopes (Meunier et al., 2008), this work presents a 2D analytical model based on a basic hydrological model applied to a hillslope and a mechanistic safety factor to evaluate atmospheric and rainfall effects on slope stability. We use the model in this paper to investigate the role of pore pressure changes induced by rainfall and atmospheric pressure changes during major storms on slope stability, while accounting for groundwater level, pre-conditioned by seasonal rainfall and compare it to the rainfall forcing.

First, we define a slope stability model based on a classical Mohr-Coulomb criterion. As both rainfall and atmospheric effects imply pore pressure diffusion in groundwater, defining slope stability requires a model able to describe groundwater diffusion. We therefore define an analytical solution for groundwater flow in a finite hillslope, and accordingly apply infiltration and

atmospheric induced pore pressures to compute slope stability changes. Second, we consider simple synthetic scenarios of
pressure and rainfall changes to model their distinct contributions to slope stability. This allows us to define spatial domains
along the hillslope where the instability is predominantly driven by either rainfall or atmospheric pressure changes. Third, we
apply this model to observed meteorological data from Taiwan to compute the respective impact of different typhoons, through
rainfall or atmospheric pressure change, on slope stability. Last, we discuss the results and the relevance of the model.

## 2 Method

### 2.1 Landslide failure mechanics

Locally, slope stability can be expressed as the stability of an infinite homogenous slope tilted with an angle $\alpha$ from the
horizontal. In the following model, a landslide occurs when a rupture happens on a slip surface (i.e., the rupture plane) that we
impose to be parallel to the topographic slope. The modelled landslide is comparable to a rigid slab sliding over a tilted surface
of the same material. The gravitational force pulls the material down and imposes a normal $\sigma_n$ and shear $\tau$ stress along the
rupture plane. We consider here that the rupture occurs if the shear stress overcomes the Mohr-Coulomb criterion:

$$\tau_c = c + \sigma_{n\,eff}\,tan\,\varphi \tag{1}$$

where $\tau_c$ is the critical shear stress, which depends on cohesion $c\ [kPa]$, the angle of internal friction $\varphi\ [°]$, and the effective
normal stress $\sigma_{n\,eff}\ [kPa]$. Most landslide analysis use a safety factor (Iverson, 2000; Hack et al., 2007; Schulz et al., 2009;
Muntohar and Liao, 2010) as indicator of slope stability. This safety factor $F$ is defined as the ratio of stabilizing forces over
destabilizing forces, i.e. the ratio of the critical shear stress over the actual shear stress:

$$F = \frac{c + \sigma_{n\,eff}\,tan\,\varphi}{\tau} \tag{2}$$

The slope reaches a critical equilibrium for $F = 1$, with any system showing a lower or a greater safety factor is considered
unstable or stable, respectively.

Slope stability can vary under the addition of external forces, or if the mechanical properties of the slope change. While
weathering processes may weaken rocks (Calcaterra and Parise, 2010; Hencher and Lee, 2010), we will focus on short-term
to seasonal processes and consider constant mechanical soil properties.  However , variations of the effective normal stress
$\sigma_{n\,eff}$ by pore pressure fluctuation is a frequent cause of slope stability change. We here define static pore pressure as the pore
pressure associated to the geometry of the water table (i.e., hydrostatic pressure) and dynamic pore pressure as the pore pressure
associated to transient effects, namely rainfall and atmospheric pressure changes. In the following, the effective normal stress
is estimated along the potential rupture plane accounting for both static and dynamic pore pressure variations induced by
rainfall and atmospheric pressure change:

$$\sigma_{n\,eff}(z,t) = \sigma_n(z) + P_a(t) - \psi_0(z) - \psi_{rain}(z,t) - \psi_{air}(z,t) \tag{3}$$

Where $\sigma_n(z)$ is the normal stress and $P_a(t)$ the atmospheric pressure at the surface. $\psi_0(z)$ is the hydrostatic component of
pore pressure, which is computed from the initial water table height. The rainfall-induced pore pressure $\psi_{rain}(z,t)$ is a

dynamic pore pressure induced by transient water table variations. These water table variations add a dynamic loading at the water table surface which then propagates downwards. $\psi_{air}(z,t)$ is the dynamic pore pressure caused by atmospheric pressure changes.

As we aim to compare these dynamic effects, the slope will be considered at yield, and only pore pressure will be investigated. In the following sections, we develop models that describe water table variations (Sect. 2.2), rainfall-induced pore pressure

$\psi_{rain}(z,t)$ (Sect.2.3) and atmospheric-induced pore pressure $\psi_{air}(z,t)$ (Sect. 2.4) during a weather event.

## 2.2 Water table model

Infinite slope models have already been developed to evaluate slope stability under rainfall forcing and the diffusion of pore pressure (eg. Iverson, 2000), but are inherently limited in groundwater flow characterisation. If recharge is the vertical movement of water, groundwater level gradients in the hillslope induce a lateral movement of water. Water table fluctuations

will change depending on the position along the hillslope, as local flow is linked to both recharge and uphill water convergence. Such characteristics cannot be represented in infinite slope models, where groundwater level is considered parallel to the surface. A more accurate description of groundwater flow is therefore required to express the flow dynamics and water table height along a hillslope.

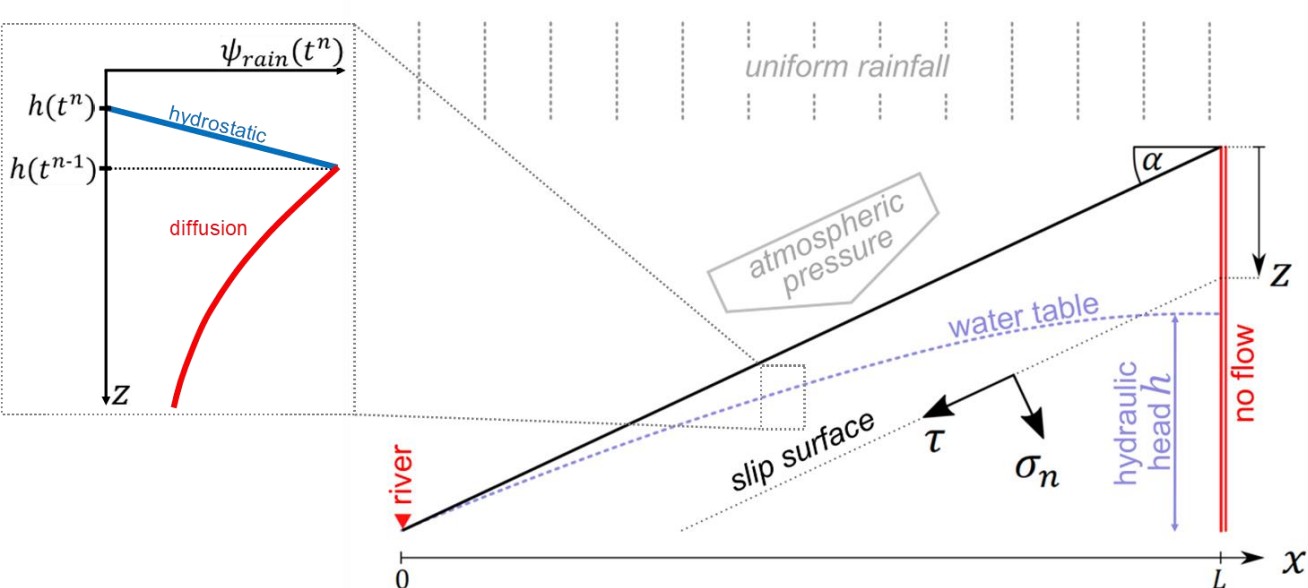

**Figure 1: Geometry of the considered hillslope. The water table (in blue) forms a quadratic surface between the two boundaries conditions (in red). The stability is evaluated with a Mohr-Coulomb criterion along a slope-parallel slip. The atmospheric pressure and rainfall infiltration are applied uniformly along the slope. The zoomed-in section shows the implementation of the diffusion of pore pressure due to the raise of water table between two consecutive timesteps.**

In the following, we develop a 2-dimensional hydrological model applied to a finite hillslope, with a slope angle $\alpha$, between $x = 0$ and $x = L$ over a deep horizontal impervious layer (Fig. 1). The water table head $h(x, t)$ $[m]$ is function of the position along the slope and depends on initial conditions and rainfall-induced recharge. Transient groundwater flow in the aquifer is described by the Boussinesq equation (Troch et al., 2013). The vertical component of flow is neglected to focus on the horizontal component (known as the Dupuit hypothesis). Furthermore, we consider that head variations are negligible with respect to the aquifer thickness, so the Boussinesq equation can be linearized as follows (Townley, 1995):

$$T \frac{\partial^2 h(x,t)}{\partial x^2} = S \frac{\partial h(x,t)}{\partial t} - R \tag{4}$$

Where a recharge $R$ $[m\ s^{-1}]$ is uniformly applied along the hillslope, $T [m^2\ s^{-1}]$ and $S$ [ ] are respectively the transmissivity and the storage coefficient of the aquifer. In the linearized form of the Boussinesq equation, T is constant and defined as the product of permeability and aquifer thickness. In an unconfined aquifer, storage coefficient $S$ is equivalent to the specific yield. Diffusivity $D$ $[m^2 s^{-1}]$ is defined as the ratio $D = T/S$.

During extreme rainfall events, groundwater recharge does not equal the amount of precipitations. Part of the rainfall will not infiltrate and generate runoff if the rainfall rate exceeds the soil infiltration capacity. This can represent a significant portion of the rainfall and is heavily dependent on soil characteristics. Therefore a limit has been set to the recharge $R$ in the form of the vertical hydraulic conductivity $K_z$ $[m. s^{-1}]$, representing the maximum capacity of the soil in terms of infiltration rate: any recharge above this level does not infiltrate and is considered as runoff.

This solution for modelling the transient water table relies on the Dupuits-Forchheimer hypothesis, with 2 assumptions. First, the flow lines are horizontal and parallel, which is verified when the lateral extend of the aquifer is much larger than its thickness, and the hillslope is not convergent nor divergent. Second, the aquifer transmissivity is not affected by water table height variations, which needs an aquifer much thicker than the amplitude of its height variations. Such hypotheses would be much suited for a long and wide hillslope with a thick saturated zone, but questionable for the steep and complex shape of hillslopes typically source of landslides, and may not exactly represent the complexity and dynamics of groundwater observed under steep hillslopes. However, it allows for a first-order and broad assessment of water table dynamics through an analytical solution, which is why it was selected.

The crest of the hillslope, $x = L$, is regarded as a groundwater divide and a Neumann no-flow condition is applied $\frac{\partial h}{\partial x}|_{x=L} = 0$, while the toe of the hillslope, $x = 0$, is considered drained by a river and the groundwater level is therefore set to the surface resulting in a Dirichlet boundary condition with $h(x = 0) = 0$.

The solution to the partial differential equation (Eq. 4) can be separated into a static part $h_s(x)$, with a constant recharge $R_s$, and a dynamic part $h_t(x, t)$, with a transient recharge $R_t$. The static solution defines a quadratic water table profile within the hillslope as a function of the distance to the hillslope toe $x$, and only depends on the length $L$ of the hillslope and the soil's hydraulic transmissivity $T$:

$$h_s(x) = \frac{R_s}{T}\left(Lx - \frac{x^2}{2}\right) \tag{5}$$

When groundwater reaches the surface, any excess of rainwater would not infiltrate but rather generate surface runoff towards the toe of the hillslope. Therefore, a hard limit has therefore been added to cap the water table at the topography and disregard any water height above the surface. Such a threshold underlines the importance of the initial groundwater level, as pore pressure can increase significantly at the crest of a hillslope while remaining nearly constant at the toe. However, this solution is not accounting for the seepage that is caused by the excess water flowing out of the soil. Seepage generates a destabilising force proportional to the flow rate, and more specifically the vertical component. Since this model assumes horizontal flow only, estimation of the seepage forces would be very inaccurate.

For the transient part of the recharge, Townley (1995) provided a solution to the equation (4) in Fourier space, describing groundwater level variations under periodic recharge. However, the weather events investigated here are not periodic, and using the solution as is would result in a partly acausal signal due to a limitation in the computation of the fast-fourier-transform algorithm. This numerical issue is avoided by considering the temporal impulse response function corresponding to Townley's solution. The transient recharge $R_t$ is convolved with this impulse function to obtain $h_t(x,t)$, the variations of the water table head as a function of time and position along the hillslope.

The hydrostatic pore pressure $\psi_0$ is then computed from the static component of the water table $h_s$, and considered as the initial state of the water table in the hillslope. The dynamic or transient fluctuation of the water table $h_t$ is a direct result of the rainfall infiltration during weather events, and describes the rise or fall of water table. These variations induce a pressure loading at the water table surface, and the propagation of this loading as pore pressure $\psi_{rain}$ is computed using a pore-pressure diffusion model (Sect. 2.3).

## 2.3 Rainfall-induced pressure diffusion

The propagation of the pore pressure induced by rainfall and water table variations can be described by a diffusion model. Iverson (2000) developed a 1D model that characterised the rainfall-induced pore pressure through a homogeneous material. While the hydrological model considers a 2D geometry, a 1D vertical model is deemed sufficient to represent pore pressure diffusion in the hillslope. Starting from Richard's equation and assuming a fully vertical diffusion and wet initial conditions, the pore pressure front $\psi_{rain}$ can be described using a one-dimensional diffusion equation:

$$\frac{\partial \psi_{rain}}{\partial t} = D \cos^2 \alpha \frac{\partial^2 \psi_{rain}}{\partial z^2} \tag{6}$$

where the maximum hydraulic diffusivity $D$ is assumed to be homogeneous, i.e. hydraulic properties do not change with depth. The characteristic time for a diffusivity equation in this context is expressed in function of the diffusion distance and the diffusivity (Iverson, 2000; Handwerger et al., 2013): $t_c = z^2/D$; and represents the minimum time at which a strong pore pressure occurs at depth $z$.

The partial differential equation (Eq. 6) is mathematically identical to the heat diffusion equation, for which Carslaw and Jaeger (1959) provided a set of analytical solutions (see chapter 2.9 of Carslaw and Jaeger, 1959). In this case, a semi-infinite solid with a Neumann condition at its surface represents well the pressure diffusion under a recharge flux at its surface. The

solution to a constant loading $H_0$ $[kPa]$ between $t = 0$ and $t = T$ is expressed using the complementary error function which is defined as $erfc(x) = 1 - \frac{2}{\sqrt{\pi}} \cdot \int_0^x e^{-z^2} dz$ :

$$\psi_{rain}(z, t \leq T) = H_0 \left[ \sqrt{\left(\frac{\widehat{D}t}{\pi}\right)} e^{-\frac{z^2}{\widehat{D}.t}} - z erfc\left(\sqrt{\frac{z^2}{\widehat{D}t}}\right) \right] \tag{7a}$$

$$\psi_{rain}(z, t > T) = \psi_{rain}(z, t \leq T) - H_0 \left[ \sqrt{\left(\frac{\widehat{D}(t-T)}{\pi}\right)} e^{-\frac{z^2}{\widehat{D}(t-T)}} - z erfc\left(\sqrt{\frac{z^2}{\widehat{D}(t-T)}}\right) \right] \tag{7b}$$

$$with\ \widehat{D} = 4D \cos^2 \alpha$$

The response to any recharge can be computed by a linear combination of these 2 solutions. Our model computes an impulse response function by replacing in equations (7) $H_0$ with the unit and taking the period $T$ equal to the time sampling. This

impulse response function can then be convolved with any recharge to obtain the associated pressure front.

The pore pressure $\psi_{rain}(z, t^n)$ is then computed using the water table variations $\Delta h_t = h_t(t^n) - h_t(t^{n-1})$ as loading (Fig. 1). The added water applies a change in weight onto the previous water table position $h_s + h_t(t^{n-1})$. The change in pressure from the added (or removed) weight of water is then used as forcing for the pore pressure model, and diffuses as pore pressure.

## 2.4 Atmospheric perturbation

Rainfall is not the only process that impacts pore pressure. As a fluid, air also contributes to pore pressure but its impact on slope stability is generally disregarded. Indeed, atmospheric pressure adds to pore pressure but also applies an equal normal load on the slope, directly increasing $\sigma_{n\ eff}$ (Eq. 3). Thus, static atmospheric pressure can be neglected as its overall effect is null. However, the variations of atmospheric pressure can have an impact on slope stability (Schulz et al., 2009). This theory has not yet been tested against natural catastrophic landslides, only on a slow-moving landslide. We therefore make the

assumption that this theory also applies for catastrophic landslides as the failure mechanisms and stability criterion are identical than for slow-moving landslides (Iverson, 2000). When an atmospheric pressure change $P_a$ occurs, it is instantaneously transferred at the slip surface as a normal stress through the assumed elastic skeleton. $P_a$ is also applied on the water table, so that the dynamic pore pressure $\psi_{air}$ adjusts by diffusion, which is a much slower process. This delay leads to a transient difference between air-induced normal stress $P_a$ and air-induced pore pressure $\psi_{air}$, changing the expression of the effective

normal stress (Eq. 3). If atmospheric pressure increases or decreases, the safety factor transiently increases or decreases, respectively.

As air is a low-viscosity fluid, pressure diffusion of the air through the unsaturated zone is considered quick enough that atmospheric pressure variations can be directly applied to the top of the water table. Diffusion process is therefore the same as for rainfall infiltration (Eqs. 7), with a Dirichlet boundary condition at the top of the semi-infinite solid instead of Neumann

one (see chapter 2.5 of Carslaw and Jaeger, 1959). The pressure input equals $P_a$ for $t \in [0, T]$ and is null otherwise.

$$\psi_{air}(z, t \leq T) = P_a erfc\left(\frac{z}{\sqrt{4Dt}}\right) \tag{8a}$$

$$\psi_{air}(z, t > T) = \psi_{air}(z, t \leq T) - P_a erfc\left(\frac{z}{\sqrt{4D(t-T)}}\right) \tag{8b}$$

As for rainfall-induced pore pressure, a numerical impulse response function is computed by taking the time sampling for $T$ and $P_a = 1$. The pressure front from any atmospheric perturbation can then be computed through a convolution between the atmospheric pressure data and the impulse response function. An effective atmospheric-induced pore pressure, noted $\psi'_{air} = \psi_{air} - P_a$, is used to compare dynamic stability changes from rainfall and atmospheric effects.

## 3 Results – Synthetic tests

The response of atmospheric and rainfall induced pore pressures to a weather event are assessed both at the toe and the crest of a modelled hillslope. For the purpose of this study, the slope is considered at yield, near the failure. The finite slope model considers a $L = 500$ m long hillslope with an angle $\alpha = 25°$, and a homogeneous cohesive soil. The soil's hydraulic conductivity has been set to $K_z = 10^{-6}\ m.s^{-1}$ as it is representative of clay soils found in Taiwan (Lin and Cheng, 2016), where we focus our study in section 4. We consider different values for the hydraulic diffusivity of $10^{-2}$, $10^{-4}$ and $10^{-6}\ m^2.s^{-1}$ to account for the large variability of natural hillslopes and a specific yield $S = 10^{-2}$. The model is first tested with synthetic inputs to characterise the changes in stability induced by rainfall and atmospheric pressure change during a simplified storm. We consider an input gate-function shape lasting 24 h to mimic a weather event (Fig. 2), even if natural signals are generally more complex. The rainfall infiltration is set equal to $K_z$ during the event and zero otherwise, which corresponds to 86.4 mm accumulated rainfall in a day. The atmospheric pressure is set to -1 kPa during the same 24 h period, and zero otherwise. We will focus on the dynamic pore pressure terms, $\psi_{rain}$ and $\psi'_{air}$, as those are the only parameters that will modify the safety factor and lead to an instability. In the following, we assess their temporal change at 5 m below the initial water table elevation, as effects are decreasing with depth (Fig. A1).

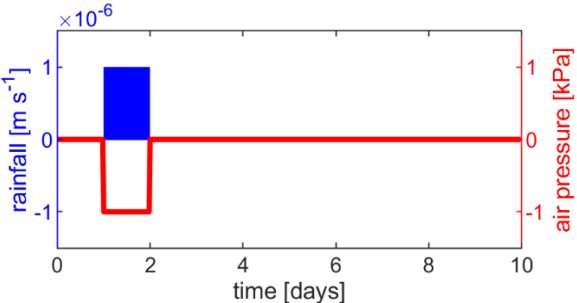

**Figure 2: Rainfall recharge and atmospheric pressure variations used for the synthetic tests. The 24 h event corresponds to a cumulated rainfall of 86.4 mm, during which the atmospheric pressure drops 1 kPa.**

Rainfall-induced pore pressure change $\psi_{rain}$ reaches its maximum after a time delay (Fig. 3a, b and c), which increases as diffusivity decreases. This delay is also function of depth (Fig. A1). However, the intensity and delay of this peak greatly depends on the diffusivity and the position along the hillslope. For a diffusivity of $10^{-2}\ m^2\ s^{-1}$ the maximum is reached in

less than 5 h after the event at the toe of the slope (at $x = 50\ m$), but it takes 17 days at the crest of the slope (at $x = 500\ m$) although the characteristic time $t_c$ is much shorter, about 41 minutes. Such a difference can be explained by the fact that $t_c$ corresponds to the time when 48% of the surface amplitude is felt at a given depth (Handwerger et al., 2013), not necessarily the maximum pore pressure value. Furthermore, the characteristic time does not consider the horizontal flow of the hillslope in its calculation. It is, however, still used as a rough approximation of the diffusion time. For such a high diffusivity, $\psi_{rain}$ shows greater values at the crest of the slope, reaching over 40 kPa, against less than 31 kPa at the toe. However, the trend is reversed for a lower diffusivity $D = 10^{-4}\ m^2\ s^{-1}$, where greater pore pressures are achieved at the toe of the slope. As for very low diffusivities, $i.e.\ D = 10^{-6}\ m^2\ s^{-1}$, no significant pore pressure response is visible in a 10 days period, as both water table variation and pore pressure diffusion are slower, with a characteristic diffusion time $t_c$ of nearly 290 days.

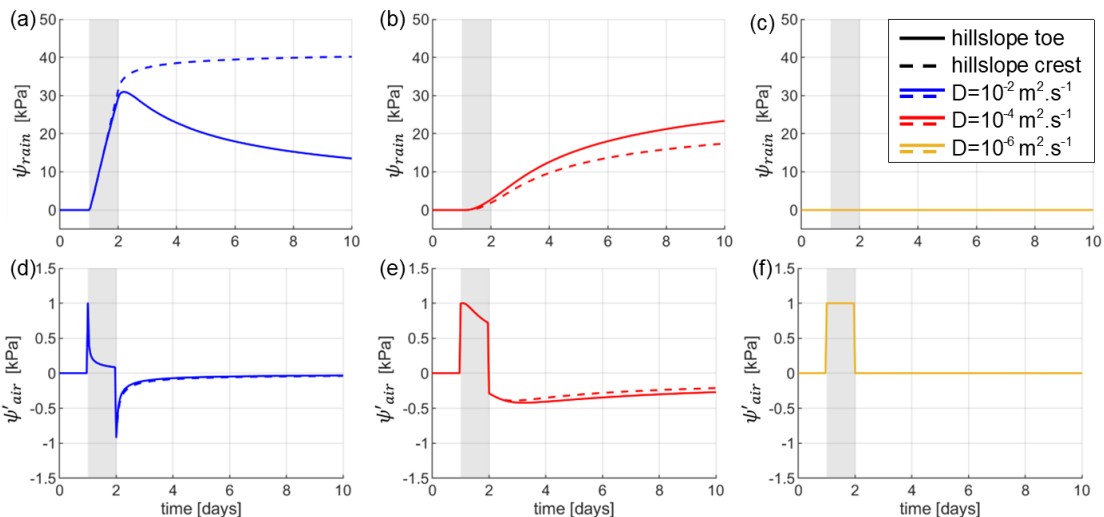

**Figure 3: Temporal evolution of $\psi_{rain}$ (a), (b) & (c) and $\psi'_{air}$ (d), (e) & (f) in response to the synthetic forcing (Fig. 2), at several diffusivities. Solid lines represent atmospheric and rainfall effects at the toe of the hillslope, at $x = 50\ m$, and dashed lines the effect at the crest of the hillslope at $x = 500\ m$. Note the difference of scale on the y-axis for a, b and c compared to d, e and f.**

The atmospheric pore pressure disequilibrium $\psi'_{air}$ shows a significantly different comportment from rainfall effects. No matter the depth investigated or hydraulic diffusivity, the maximum response to atmospheric pressure drop shows no delay and is always equal to the inverse of the pressure change (Fig. 3d, e and f). However, the higher the hydraulic diffusivity, the faster $\psi'_{air}$ returns to a value of zero. This means that the effect is short for shallow or diffusive media but lasts during the full depression for deep or low diffusivity media. The negative atmospheric pressure change at $t = 24\ h$ leads to a 1 kPa positive peak in effective pore pressure $\psi'_{air}$. Similarly, at the end of the event, the atmospheric pressure increase causes a 1 kPa decrease of $\psi'_{air}$, stabilising the hillslope after the event.

The slight discrepancy between $\psi'_{air}$ at the crest and the toe of the hillslope is due to the greater water table rise during the event, which leads to a greater diffusion distance.

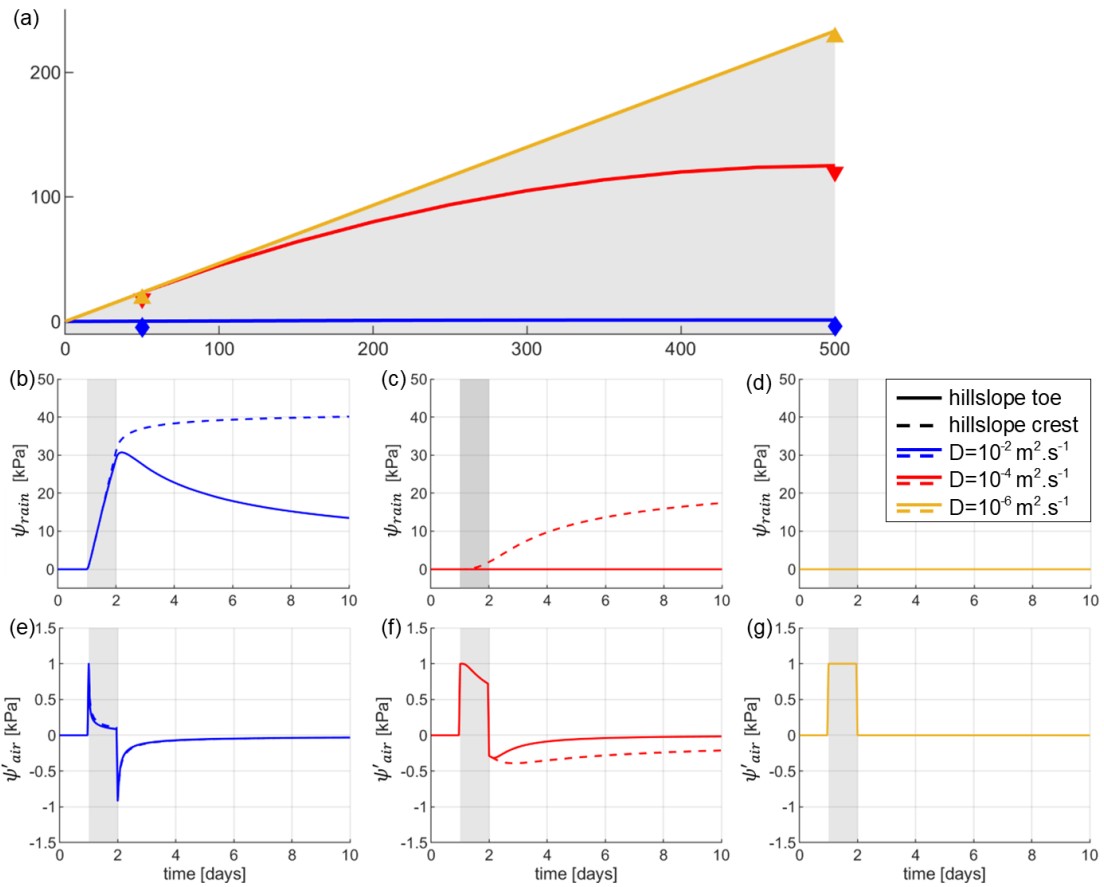

Figure 4: Initial state of the water table in the hillslope (a) after a static recharge of $10^{-9}\ m\ s^{-1}$ and investigated points at the crest and toe of the hillslope. Temporal evolution of $\psi_{rain}$ (b), (c) & (d) and $\psi'_{air}$ (e), (f) & (g) in response to the synthetic forcing (Fig. 2), at several diffusivities. Solid lines represent atmospheric and rainfall effects at the toe of the hillslope, at $x = 50\ m$, and dashed lines the effect at the crest of the hillslope at $x = 500\ m$.

We now consider in Fig. 4 the role of the initial water table height on the impact of rainfall and atmospheric pressure change on slope stability. During the event, the same input functions are used as for the previous case (Fig. 2), but a constant recharge of $10^{-9}\ m.\,s^{-1}$ is added prior and after the event, corresponding to 2.6 mm in a month. Adding even a slight static recharge drastically changes the initial water table height prior to the weather event (Fig. 4a). We here mostly focus on the impact of the initial water table height on $\psi_{rain}$, as $\psi'_{air}$ is not expected to change significantly with the initial conditions of the water table. We find that the impact of the initial water table height on $\psi_{rain}$ strongly depends on diffusivity. For a low diffusivity of $10^{-6}\ m^2\ s^{-1}$, the hillslope is already fully saturated, and no further infiltration can occur (Fig. 4d). In that case, the absence of pore pressure is not related to a slow response due to low diffusivity, but to the lack of rainfall infiltration. For higher diffusivities or lower constant recharge, saturation occurs systematically at the toe of the hillslope, where the water table is the closest from the topography. For $D = 10^{-4}\ m^2\ s^{-1}$ (Fig. 4c), the response of $\psi_{rain}$ at the crest of the hillslope is similar to the one without any static recharge, while near the toe of the hillslope, the water table reaches the surface and $\psi_{rain}$ shows no

effect. For a high diffusivity of $10^{-2}\ m^2\ s^{-1}$ (Fig. 4b), the change in the initial water table height due to the static recharge is limited and does not lead to strong differences in temporal changes of $\psi_{rain}$.

If the initial water table height does not significantly impact $\psi'_{air}$, it leads to variations in terms of the dominant cause of instability between $\psi_{rain}$ and $\psi'_{air}$ along the hillslope. Indeed, the stability of the already saturated hillslopes, prior to the weather event, can only be reduced by changes in $\psi'_{air}$, even if the amplitudes of these changes remain limited to 1 kPa. This

occurs everywhere along the hillslope for $D = 10^{-6}\ m^2\ s^{-1}$ , while only the hillslope toe is dominated by $\psi'_{air}$ for $D = 10^{-4}\ m^2\ s^{-1}$. For $D = 10^{-2}\ m^2\ s^{-1}$, changes in $\psi_{rain}$, up to ~40 kPa, overcome $\psi'_{air}$ by more than order of magnitude – except for the very beginning of the event. This suggests that atmospheric effect should be the dominant factor only in the already saturated part of the hillslope, such as close to the toe, where rainfall induced dynamic pore pressure change is null or low, or at very short timescales, since it is instantaneous.

## 4 Results - Application to natural datasets

### 4.1 Data set

Taiwan is a mountainous island coming from the convergence between the Eurasian and the Philippines plate. A large portion of the island is composed of steep slopes and mountains, which culminates at 3952 m above sea level. The relief are very steep and composed of sandstone, slate, schist and mudstone (Lin et al., 2011; Tsou et al., 2011). However, a large portion of the

285 surface material are very significantly weathered due to the annual precipitation of 2.5 m. As a region undergoing several typhoons each year and subjected to landslides, Taiwan is a relevant study area.

Weather data were obtained from the Data Bank for Atmospheric Research at the Taiwan Typhoon and Floods Research Institute. The data is an hourly report of rainfall and atmospheric pressure, from 01/01/2003 to 30/06/2017. The weather station is located in the Taroko national park, in north-eastern Taiwan (C0U650, 24.6753 lat, 121.5871 long).

In the model, the recharge is assumed to be equal to the observed rainfall, neglecting evapotranspiration. Atmospheric tides are observed in the atmospheric pressure data, with a diurnal and semidiurnal period and amplitudes of about 0.03 to 0.1 kPa respectively. These tides are removed using notch filters to focus only on typhoons. In a similar way, a high pass filter is applied to only keep signals with a period less than 30 days and remove seasonal components. In the following, we assume that any remaining change in atmospheric pressure is attributable to weather events.

A total of 36 major typhoons are identified in the data. Rainfall peak intensity ranges roughly between 0 and 57.6 mm.h$^{-1}$, and atmospheric pressure drop reaches down to -4.5 kPa. Among the 36 major typhons, some have led to a strong pressure drop and/or to intense rainfall (Fig. 5a). For this study, 3 contrasting typhoons are used to compare atmospheric and rainfall effects: Matsa, Krosa and Morakot. Typhoon Masta in July 2005 is the event showing the highest peak of rainfall intensity among the dataset. Typhon Masta led to several mudslides and floods in Taiwan, but no major landslide. Typhoon Krosa in October 2007

is associated with the highest atmospheric depression in the dataset. It also resulted in minor damages as it passed directly over

the island of Taiwan. Typhoon Morakot in August 2009 was devastating and caused more than 10,000 landslides (Lin et al., 2011; Lin and Lin, 2015; Hung et al., 2018; Steer et al., 2020), including the Shiaolin landslide which mobilized a volume of $25 \times 10^6$ m³ and buried the village of Shiaolin (Tsou et al., 2011; Kuo et al., 2013). We highlight here that Typhoon Morakot was associated with a moderate pressure drop and peak rainfall intensity at the location of the weather station in Taroko, but led to extreme rainfall intensity in southern Taiwan reaching close to 100 mm h⁻¹ (Yang et al., 2018).

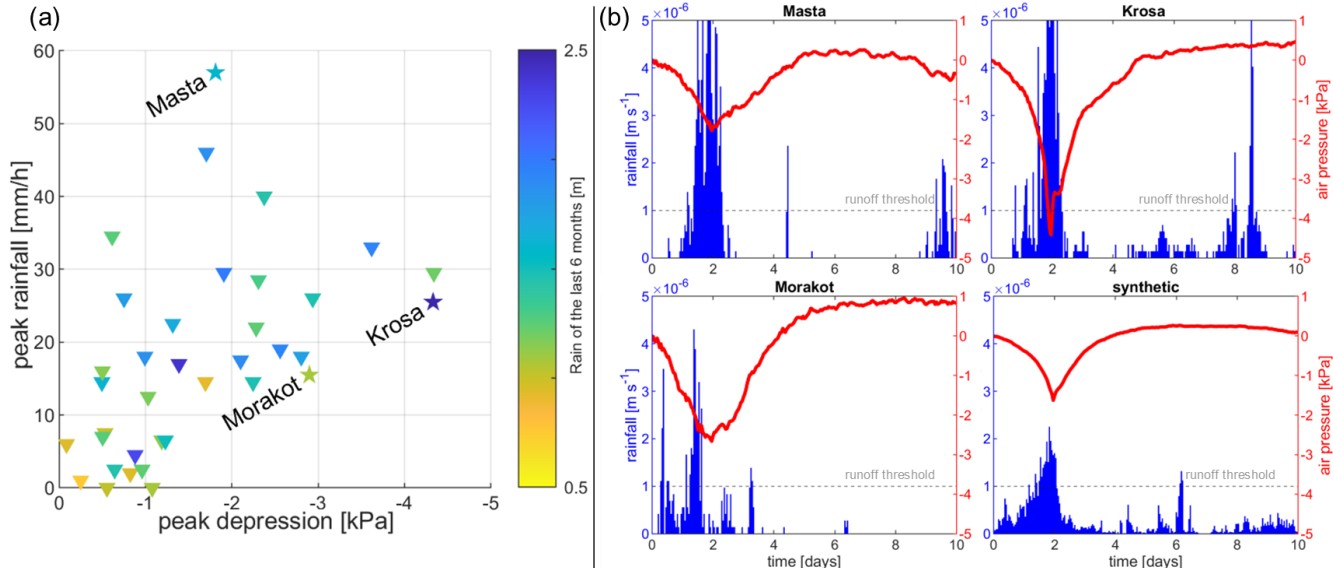

**Figure 5: (a) Typhoons over Taiwan sorted in function of their maximum rainfall intensity and atmospheric pressure drop. The hydrological context is represented by the colorscale, showing the cumulated rainfall over the 6 months prior the event. (b) Timeseries of rainfall and atmospheric pressure changes of the 3 typhoons and the synthetic event created from the average of every typhoon.**

On top of these three events, a theoretical typhoon is tested by taking the arithmetic mean of rainfall and atmospheric pressure of all 36 events in the data (Fig. 5b). The atmospheric pressure profile of this 'average typhoon' is similar to the form of the pressure cross-section of a typhoon described by the empirical Griffith model (Griffith, 1978).

### 4.2 Finite hillslope model

The impact of typhoons Masta, Krosa, Morakot and the average typhoon have been investigated through the hillslope stability model. The initial state – as previously established with the synthetic tests – plays an important role when computing $\psi_{rain}$, by constraining the water table position, and therefore the maximum for the dynamic pore pressure. To account for the hydrological context of each typhoon, the mean recharge of the 6 months before the typhoon is used to compute the initial water table level.

For most typhoons, the amount of rainfall received during the preceeding 6 months is significant, with average rates ranging between $4.9 \times 10^{-8}\ m.s^{-1}$ and $1.6 \times 10^{-7}\ m.s^{-1}$. Such a recharge greatly impacts the initial water table level. For low diffusivities such as $10^{-4}\ m^2\ s^{-1}$ and $10^{-6}\ m^2\ s^{-1}$ the hillslope is already fully saturated and its water table reaches the

topography before the typhoon occurs. This prevents any rainfall-induced pore pressure during the typhoons, leaving only the atmospheric response as a potential destabilizing factor. Even for the relatively high diffusivity of $10^{-2}$ $m^2$ $s^{-1}$, some typhoons are striking hillslopes already fully saturated at their toe and potentially above (Fig. 6a). As an example, typhoon Krosa, which occurs at the end of the typhoon season, shows the highest initial static recharge with a total of 2.51 m of precipitation in 6 months prior the actual typhoon. In turn, four fifths of the hillslope are already fully saturated. In our set of tested events, only typhoon Morakot and the synthetic mean event occur in a context where the toe of the hillslope is not fully saturated, with only 1.05 and 1.46 m respectively of cumulated rainfall during the 6 preceding months.

At the toe of the hillslope, Morakot and the synthetic mean event are the only events showing a non-zero $\psi_{rain}$ at $D = 10^{-2}$ $m^2.s^{-1}$. In the case of Morakot, rainfall-induced pore pressure rapidly peaks above 33 kPa within 3 days of the start of the rainfall (Fig. 6d), less than 2 days after the maximum atmospheric response. As for the synthetic mean event, $\psi_{rain}$ reaches its maximum load under 9 kPa in a day (Fig. 6e), which illustrates that the water table has reached the surface and that subsequent rainfall is not infiltrating.

At the crest of the hillslope, for a high diffusivity $D = 10^{-2}$ $m^2.s^{-1}$, the rainfall-induced pore pressure is exceeding 100 kPa after 10 days in some instances. However, the pore pressure increase is not faster than at the toe of the slope, and $\psi_{rain}$ is still increasing after 10 days.

The atmospheric-induced instability $\psi'_{air}$ reaches values of 0.3 to 1.5 kPa depending on the event (Fig. 6f to i), 1 or 2 orders of magnitude smaller than $\psi_{rain}$. By the time $\psi'_{air}$ reaches its maximum value, $\psi_{rain}$ already exceeds 20 kPa if the hillslope is not fully saturated. Indeed, the rainfall tends to occur just before the main atmospheric drop (Fig. 5b). The peak responses do not match the recorded atmospheric pressure drops. Indeed, Krosa shows an atmospheric pressure drop of 4.5 kPa, but $\psi'_{air}$ only reaches 1.5 kPa at $D = 10^{-2}$ $m^2.s^{-1}$ (Fig. 5a and Fig. 6g). This is because the drop of atmospheric pressure takes several hours, even days, to reach its lowest value. During this delay, the diffusion process already starts to readjust pore pressure to the atmospheric one, decreasing the overall effect. $\psi'_{air}$ is therefore more important and closer to the opposite of $P_a$ the lower the diffusivity (Fig. A2). $\psi'_{air}$ also slightly increases upslope, compared to the toe of the hillslope, with up to a 42.9 % higher response for the typhoon Krosa. This is due to the rise of the water table during the typhoon – as previously discussed in Sect. 3 – which increases the diffusion length and slows down the return to equilibrium.

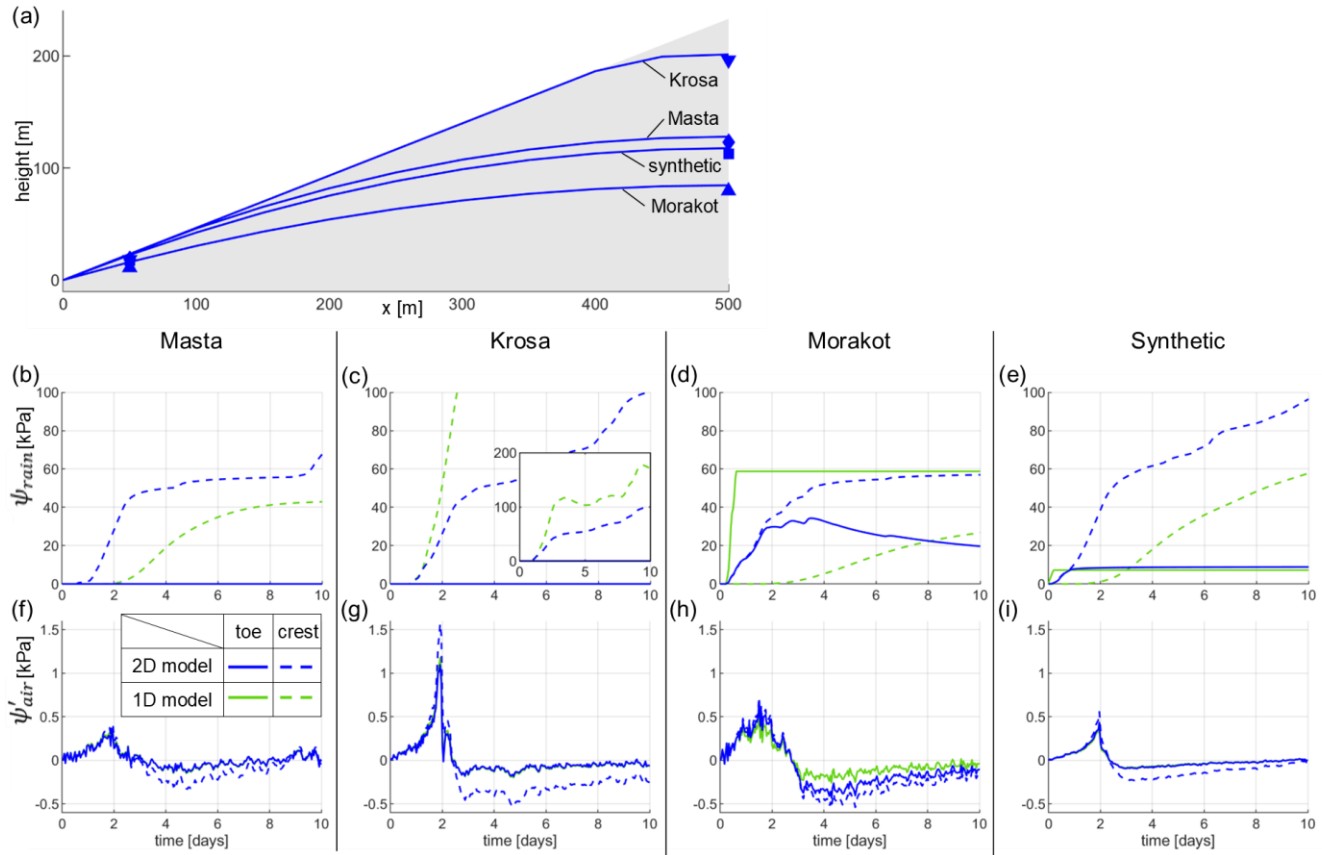

**Figure 6: Initial state of the water table in the hillslope (a) before each typhoon, for a diffusivity of $10^{-2}\ m^2.s^{-1}$. Markers indicate the location of investigated points in the fallowing figures (b-i), 5m under the initial water table both at the crest and the toe of the hillslope. The temporal evolution of $\psi'_{air}$ (b-e) and $\psi_{rain}$ (f-i) in response to the four typhoons using the hillslope model (blue lines). Solid and dashed lines represent atmospheric and rainfall effects at the toe or crest of the hillslope, respectively. The black solid and dashed lines represent the equivalent to the blue ones, computed using the 1D infinite slope model described in Sect. 5.1, at similar depths.**

# 5 Discussion

## 5.1 Model limitations

The models presented in this study consider simplification hypotheses, for both the failure mechanism and the hydrological characterisation of the slope. The finite hillslope hydrological model, which proposes a more realistic formalism for groundwater flow that the infinite slope model, allows for a simple characterization of both rainfall and atmospheric effects on slope stability along the slope. However, the finite hillslope model is based on a Dupuit hypothesis and considers small water table level variations compared to the aquifer width (Townley, 1995). Therefore, this model describing the water table is less adapted to steep hillslopes such as those found in Taiwan.

While considering the full hillslope and groundwater dynamics helps represent pore pressure diffusion and the resultant instabilities, considering a homogeneous hillslope with a single unconfined aquifer is still a simplification, which neglect the potential role of perched aquifer within the hillslope. However, the model can be applied at any scale as long as the boundary conditions and conditions and the hypothesis of the hydrological model are respected.

Another limitation of the infinite and finite hillslope models is the independent computation of rainfall-induced and atmosphere-induced pore pressure diffusion. Indeed, rainfall infiltration tends to create a downward fluid displacement, while a drop of atmospheric pressure tends to induce an upward fluid flow, as it moves from high to low pressure areas. These two mechanisms happen simultaneously during a weather event and can, in turn, interact with each other. As the model limits lateral water movement as a diffusion process, the time delay between rainfall and the hydromechanical response can be overestimated. We also consider that a fully saturated hillslope does not show any response to rainfall in terms of stability in the model. However, if the water table reaches the surface, even though the charge of the column of water does not change, the water flowing out of the slope induces a destabilising force function of the flow rate. This phenomenon, known as seepage, can lead to slope failure induced by rainfall near the toe of the hillslope (Budhu and Gobin, 1996; Ghiassian and Ghareh, 2008; Marçais et al., 2017). However, accounting for this process would require a dynamic computation of flow.

Finally, the hillslope model considers a fully homogenous material, with no mechanical properties changes along the slope nor with depth. This simplification hypothesis sets aside the complexity of the soil, especially with regards to the weathering.

## 5.2 Benefits of a groundwater finite hillslope model to assess landslide hazard

We here compare the finite hillslope model, considered in this manuscript, to a classical 1D model which considers an infinite slope and slope-parallel water table and flow (Iverson, 2000). In this 1D model, the water table is fixed, and the rainfall-induced pore pressure $\psi_{rain}$ starts diffusing from the surface, regardless of the depth of the water table. Atmospheric-induced pore pressure is however diffusing from the water table, as for the finite hillslope model. Both rainfall and atmospheric induced diffusion processes are described and computed using the same equations (Eqs. 7 & 8) in the two models. The exact same conditions as for the finite hillslope model has been applied, using the same four typhoon events. In particular, the water table in the 1D model is set to match the initial states computed in Fig. 6a for each typhoon, and atmospheric and rainfall effects are evaluated 5 m under the water table.

The main difference between the 1D infinite slope model and the finite hillslope model is the presence of a dynamic water table in this latter. Another significant difference is the point at which rainfall-induced pore pressure is applied. Indeed, the infinite slope model diffuses $\psi_{rain}$ from the surface, while the finite hillslope model converts rainfall into water table variation and directly applies the corresponding pore pressure $\psi_{rain}$ to the water table surface. This lack of infiltration model in the unsaturated part of the hillslope model prevents any shallow landslides above the water table and leads to quicker response times when the water table is deep.

The atmospheric effect $\psi'_{air}$ does not significantly vary between the finite hillslope model and the 1D infinite slope model.
The values are slightly underestimated using the latter because the water table is fixed at a certain depth and does not account for the rise of the water table, which extends the distance from which the pore pressure must diffuse through.

However, the results are significantly different for the rainfall-induced pore pressure $\psi_{rain}$. When the water table is deep (e.g., 100 m below the surface), the 1D model response is delayed and smaller than the hillslope one. For example, at the crest of the hillslope during typhoons Masta, Morakot and the synthetic event (Fig. 6), the $\psi_{rain}$ response from the 1D model starts 1 to 2 days later than when using the finite hillslope model and reaches values 36 to 53 % smaller after 10 days. This difference occurs because pore pressure diffusion starts at the surface for the infinite model and not at the water table surface as in the finite hillslope model, leading to an increased diffusion distance in the infinite model and in turn to a delayed and reduced response. On the other hand, the finite hillslope model lacks an infiltration model, and the rainfall is entirely and immediately converted in water table variations, which might underestimate the response time. When the water table is shallow (e.g., 32 m under the surface or less), $\psi_{rain}$ increases faster and reaches greater values in the infinite model than in the finite hillslope model, because in the latter groundwater flow drains part of the recharge towards the river. This is the case for the crest of the hillslope during typhoon Krosa and the toe of the hillslope during Morakot or the synthetic event (Fig. 6).

These differences between the 1D and finite hillslope models can lead to significant changes when applied to specific typhoons, with large implications for hazard assessment. For example, these two models lead to stark difference for $\psi_{rain}$ during and after typhoon Morakot, which was the source of more than 10.000 landslides. The 1D infinite slope model predicts a rapid step-like increase in $\psi_{rain}$ during the first day of the typhoon, while the finite hillslope model predicts a smoother increase peaking during the third day of the typhoon. This could explain the timing of some landslides triggered during typhoon Morakot, as for instance the catastrophic Shiaolin landslide which occurred during the third day of typhoon Morakot and led to major damages and numerous casualties (Kuo et al., 2013). At a first order, the finite hillslope model seems to be more relevant for estimating the slope stability and the timing of this specific failure since it is synchronous with the maximum value of $\psi_{rain}$. The lack of representation of lateral groundwater flow in the 1D infinite slope model may lead to a large overestimation of the rainfall effect, especially near the boundaries (water divide or river). Considering the hydrological evolution and dynamics of the full hillslope likely allows for a better estimation of $\psi_{rain}$.

**5.3 Model sensitivity to hydrologic diffusivity**

Pore pressure changes induced by rainfall and atmospheric pressure changes are both diffusive mechanisms (Eqs. 7 & 8) and are both sensitive to hydraulic diffusivity. Hydraulic diffusivity is highly variable in space, and its estimation is complex and scale dependent (Jiménez-Martínez et al., 2013). As an example, measurements can vary over several orders of magnitude inside a single slope, and the scale of the hillslope or the presence of preferential flowpaths may lead to biased values and overestimation of the diffusivity (Handwerger et al., 2013). When focusing on soils, hydraulic diffusivities are typically low, ranging between $10^{-2}$ and $10^{-7}$ $m^2\ s^{-1}$ (Reid, 1994; Iverson, 2000; Chien-Yuan et al., 2005; Baum et al., 2010; Berti and Simoni, 2012; Handwerger et al., 2013; Finnegan et al., 2021). These values are more adapted to clayey and silty soils and

corresponds to the type of soils found on the hillslopes in Taiwan (Lin and Cheng, 2016). In groundwater studies, diffusivities are larger, typically ranging between $10^{-2}$ and $10^2 \ m^2 \ s^{-1}$ (Jiménez-Martínez et al., 2013), with large values in highly fractured systems and some specific sandy aquifers in Taiwan – as high as $3.5 \times 10^2 \ m^2 \ s^{-1}$ (Shih and Lin, 2004; Knudby and Carrera, 2006). Moreover, effective hillslope diffusivity varies as a function of the saturation level of the soil above the water table, and is therefore likely to vary throughout the year and the seasons (Finnegan et al., 2021).

Media hydraulic diffusivity is a key factor controlling pore pressure and its effect on slope stability. The higher the diffusivity, the greater will be the impact of rainfall (Fig. 4). The atmospheric effect is also affected by diffusivity. The higher the diffusivity, the faster pore pressure readjusts to the atmosphere and the quicker $\psi'_{air}$ decreases. In case of a discontinuous gate function for atmospheric pressure (Fig. 2), the maximum value reached by $\psi'_{air}$ is not affected by a change of diffusivity. On the other hand, when considering real continuous data, where the atmospheric pressure takes a couple of days to reach its lowest value, the maximum of $\psi'_{air}$ decreases with increasing diffusivity (Fig. 6f to i and Fig. A2), because the readjustment process has already started by the time the peak is reached. $\psi_{rain}$ and $\psi'_{air}$ are impacted by the diffusivity in opposite ways – a low diffusivity favours rainfall-induced pore pressure, and a high diffusivity favours atmospheric-induced pore pressure; therefore the diffusivity has a great impact on the driving mechanism for failure.

The water table is also diffusivity-dependant (Eq. 4), for both its static level and its variations. The static level is inversely proportional to the hydraulic diffusivity (Eq. 5), so a decrease in diffusivity will result in increasing water table height. A low-diffusivity hillslope is therefore more susceptible to be initially fully saturated by the mean recharge of the previous months, nullifying the dynamic effect of rainfall $\psi_{rain}$. At the contrary, greater dynamic variations of water table are achieved for greater diffusivities, leading to greater pore pressure response $\psi_{rain}$. Overall, high-diffusivity slopes will be more susceptible to rainfall effects, whereas low-diffusivity hillslopes are likely to be fully saturated and, in turn, to be destabilized by atmospheric pressure changes.

**5.4 Respective role of rainfall and atmospheric effects on pore pressure changes and slope stability**

Even though rainfall-induced pore pressure and atmospheric effects are both based on the same diffusivity mechanism, their impact on slope stability are very different. $\psi_{rain}$ is a pore pressure diffusion in response to a change in water table height. Pore pressure will diffuse slowly downwards in function of soil diffusivity, and the deeper, the smaller the change in pore pressure. On the other hand, the effective-atmospheric induced pore pressure results from the difference between the atmosphere pressing on the hillslope and the pore pressure diffusion readjusting to the new value. Therefore, $\psi'_{air}$ response to an atmospheric pressure drop is instantaneous and does not decrease with depth. On the contrary, it is reinforced with depth as the diffusion process will have to go through more material before readjusting the pore pressure.

While $\psi_{rain}$ decreases with depth and $\psi'_{air}$ increases up to reaching the opposite of atmospheric pressure variations, both of their relative effects on slope stability tends towards zero deep under the surface. Indeed, stresses $\sigma_n$ and $\tau$ increase linearly with depth, so that a net decrease of the effective normal stress will have a neglectable impact in a high stress environment

(Eq. 2). Because of its small values, around 1 kPa, $\psi'_{air}$ is not expected to greatly impact slope at great depths, but rather to induce instabilities at shallow depths in the saturated part at the toe of a hillslope. The rainfall effect, however, can reach values of tens of kPa or even higher near the water table (Fig. 6), under high diffusivity conditions. According to the model, landslides triggered by intense rainfall events on already partially saturated slopes are more likely to occur just beneath the water table, in the upper part of the hillslope even if the water table is deep under the topography in this location.

**5.5 The location of landslides triggered by typhoons occurring after a wet or a dry season**

The geomorphological and hydrogeological context of the location considered plays an important role when assessing slope stability. For instance, the position along the hillslope has a great influence on the dynamic of the water table. Indeed, water table variation depends on the boundaries of the hillslope, namely the water divide and outlet. The position along the hillslope of the maximum variations of the water table is function of the diffusivity, but also the length of the slope and the period of the rainfall recharge (Townley, 1995). Water table variations tend to reach a maximum near the crest of the hillslope when recharged by intense rainfall events, as typhoons. This implies higher values of $\psi_{rain}$ near the crest of the hillslope (Fig. 6). On the other hand, the presence of the river imposing a Dirichlet condition at the toe of the hillslope, forces the water table variations to be zero at $x = 0$. However, even near the toe of the hillslope, at $x = 50\,m$, rainfall effects are still one order of magnitude greater than atmospheric ones, provided the hillslope is not initially fully saturated. Indeed, $\psi'_{air}$ is barely affected by the position along the hillslope, with a slight increase of the effect towards the crest of the hillslope, where a greater water table rise increases the diffusion distance.

The initial elevation of the water table constrains the maximum amplitude of the rain-induced pore pressure. Typhoons Krosa and Masta occurred at a state where the modelled water table reached the surface at the toe of the slope, for a high diffusivity $D = 10^{-2}\,m^2.s^{-1}$, preventing any further rise of the water table or any increase in $\psi_{rain}$ (Fig. 6a). Typhoon Krosa occurred at the very end of the wet season, in early October 2007, after 2.5 m of cumulated rainfall during the past 6 months. The modelled hillslope is saturated up to the four fifths just prior to the typhoon, restricting the rise in pore pressure to the crest of the hillslope. Typhoon Morakot occurred after a relatively dry period, with some areas reporting no rainfall during the 2 months prior the event (Kuo et al., 2013). Hence, the modelled initial water table lies over than 7 m below the surface at the toe of the hillslope, potentially enabling the rainfall effect $\psi_{rain}$ to increase pore pressure by more than 30 kPa. In the case of the synthetic mean typhoon, the modelled initial water table lies only 1 m under the surface at the toe of the hillslope. Saturation is therefore rapidly reached during the event and pore pressure increase caps off at ~9 kPa while the water table rises (Fig. 6e).

Generally, towards the crest of the hillslope, where the hillslope is not fully saturated, rainfall effects are dominant (Fig. 7). Downslope, below the point where the water table reaches the topography, atmospheric effects are potentially dominant since they are the only dynamic effects. The limit between the atmospheric-driven domain and the rainfall-driven one will shift along the hillslope in function of the past rainfall and initial height of the water table. In a wet season, where most of the hillslope is fully saturated, the limit shifts upwards, promoting atmospheric effects, while in a dry context, the limit shifts downwards, promoting rainfall effects.

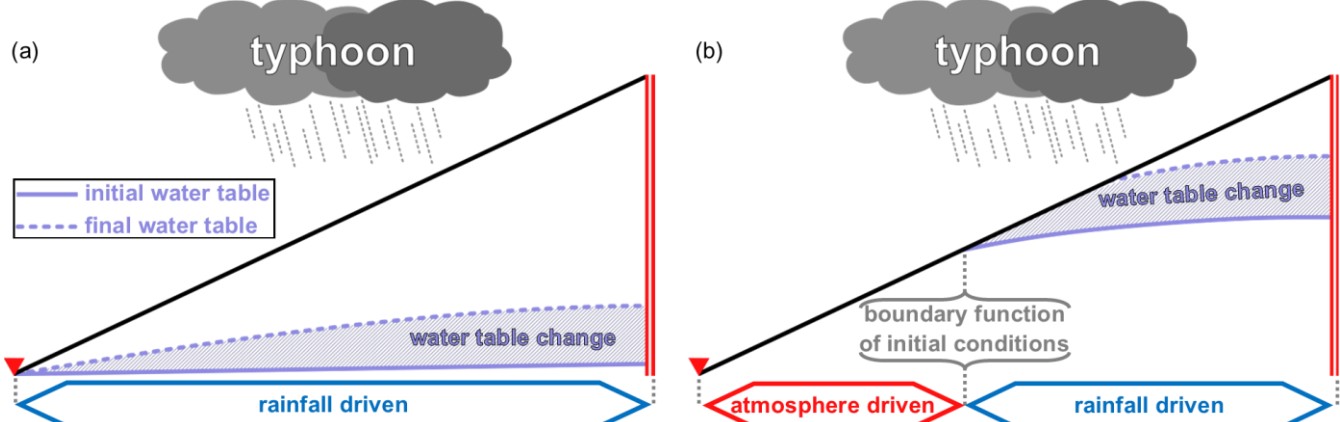

**Figure 7: Diagram representing the hillslope and the main driving effect for a potential landslide during a typhoon. If the water table is deep (a), because of a high diffusivity or dry season, the rainfall effects are dominants all along the hillslope. However, if the initial water table reaches the surface (b), because of a low diffusivity or a wet season, any failure near the toe of the hillslope will be driven by atmospheric effects only. The boundary between atmospheric driven and rainfall driven domain shifts towards the crest of the hillslope the higher the initial water table.**

The geometry of the hillslope controls this distribution as well: the water table is less likely to reach the topography in a very steep and highly diffusive hillslope than in a shallow low diffusivity hillslope. Moreover, the shape of the hillslope also plays an important role when determining the water table profile. In this case, the model assumes a hillslope of constant angle and width, water divide and outlet lines of same lengths. However, a converging or diverging topography will change the drainage area and the steady state of the water table (Troch et al., 2002; Marçais et al., 2017). Converging topography will increase the saturation near the toe of the slope, while a diverging one will have the opposite effect.

This non-uniform distribution of the destabilizing mechanisms along the hillslope suggests a non-uniform distribution of landslides triggered by weather events. This in accordance with observations of landslides distribution in Taiwan, where typhoon-induced landslides were found to occur close to the toe of hillslopes, in contrast with the relatively uniform distribution of earthquake-induced landslides along hillslopes (Meunier et al., 2008). Therefore landslides triggered by typhoons tend to occur in the atmosphere-driven zone (Fig. 7), suggesting they occur due to atmospheric pressure changes. No direct conclusions should be drawn however, as other phenomenon can explain this distribution. This study focuses on the dynamic effects on $\sigma_{n\,eff}$, computing $\psi_{rain}$ and $\psi'_{air}$ at a fixed depth under the water table. But the water table itself is closer to the surface at the toe of the hillslope leading to greater hydrostatic pore pressure $\psi_0$ and decreasing the safety factor. Another valid explanation to this landslides distribution is the effect of seepage at the toe of hillslopes, where groundwater can flow upwards, leading to soil liquefaction at high flow rates.

### 5.6 Timing of the failure during an extreme weather event

Most datasets on landslides occurring during a triggering event are based on comparison between pre- and post-event satellite images (Cheng et al., 2004; Nichol and Wong, 2005; Martha et al., 2015) or even Lidar data (Bernard et al., 2020), often

acquired days or weeks apart. The timing of landslide occurrence during the event itself remains poorly constrained. This is problematic when trying to attribute landslides to their triggering factor, whether rainfall or atmospheric pressure drop. It results that most landslides are by default attributed to rainfall. At first order, this is a reasonable assumption given that it is

520 the effect leading to the greatest disturbances compared to atmospheric effects. However, this prevents a better understanding of landslide triggering during storms, as $\psi_{rain}$ and $\psi'_{air}$ behave differently, with potential implication for landslide hazard. Based on our modelling results, we therefore give in the following some first order criterion to distinguish landslides triggered by rainfall or by atmospheric pressure drop.

The rainfall-induced pore pressure follows a diffusion mechanism and is delayed from the rainfall infiltration in function of

525 diffusivity and depth (Fig. 4). The response time of the water table (Eq. 4) can be approximated to the first order by $t_c$, even though this equation has been found to be unprecise when estimating response times (Handwerger et al., 2013) – for example underestimating the peak of $\psi_{rain}$ as seen in Sect. 3, or as depicted in Fig. A1. The time to the peak response of rainfall-induced pore pressure also changes with the position along the hillslope. Downslope, the proximity of the river – represented by a Dirichlet boundary condition – prevents significant water table variations and drains groundwater. This induces a smaller

response and a swift decrease of $\psi_{rain}$ in the lower part of the slope. Rainfall can be expected to trigger landslides within a few hours or days (depending on the diffusivity and depth of the sliding surface) at the toe of the hillslope. On the other hand, near the crest of the hillslope, $\psi_{rain}$ reaches higher values but peaks after a significantly longer time. As an example, for a diffusivity of $10^{-2} \ m^2 \ s^{-1}$, $\psi_{rain}$ reaches over 30 kPa in less than 5 h at the toe of the slope ($x = 50 \ m$), and over 40 kPa 17 days after the end of the event at the crest of the slope ($x = 500 \ m$) (Fig. 4). Therefore, our model suggests rainfall-induced

landslides might be susceptible to occur up to several weeks – or even months depending on the diffusivity – after the rainfall event.

On the contrary, the atmospheric effect $\psi'_{air}$ on slope stability is instantaneous and applies anywhere under the water table. Atmospheric-induced landslides are therefore susceptible to occur during the depression, while air pressure decreases or is at its lowest point. This corresponds to an early stage during the typhoon event, in phase with the peak of rainfall (Fig. 6) and

540 would lead to the early failure of slopes close to yield. It also means that atmospheric depressions, not associated to significant rainfall, could potentially trigger landslides on the least stable hillslopes, leading to a limited number of landslides at a regional scale.

Hillslope's length also affects the timing of the response. Indeed, the length $L$ between the upper and lower boundary condition affects the water table response. A smaller hillslope would produce a similar water table profile than the one presented in this

paper, yet with a faster response, following the scaling of the maximum characteristic horizontal diffusion time $L^2/D$. The quadratic length coefficient and the very wide range of diffusivity lead to a wide range of response time, from hours to years, depending on the hillslope properties (Fig. 8).

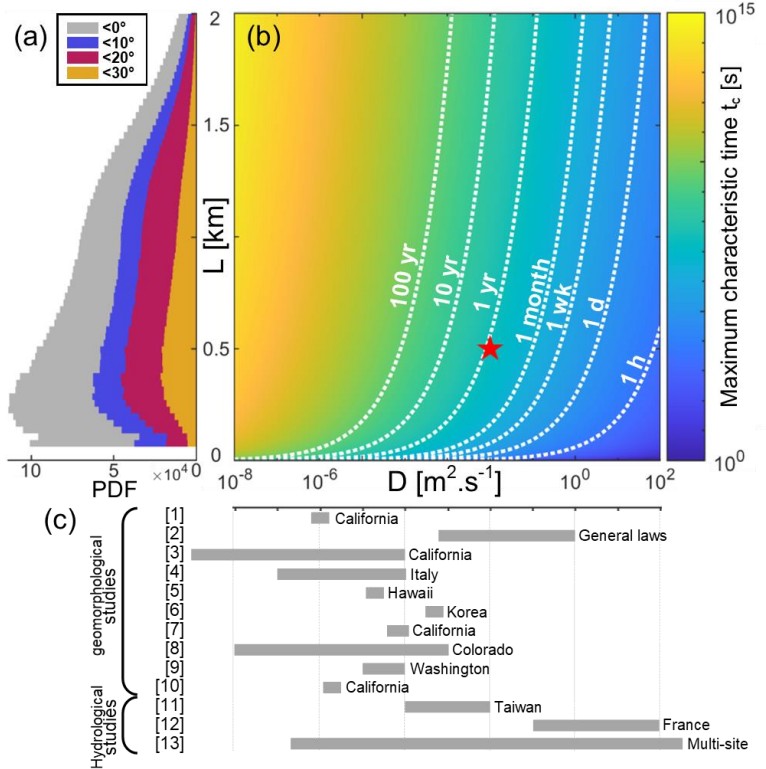

**Figure 8: Maximum characteristic timescale of hillslopes response depending on diffusivity D and hillslope length L. The histogram (a) shows the repartition of hillslopes lengths in Taiwan, extracted from a $30 \times 30$ m DEM. The maximum characteristic hillslope response time $L^2/D$ is presented in (b), with the point corresponding to the values used in this study highlighted. Values of diffusivity found in literature are displayed in (c). Sources from the diffusivity graph (c) are: [1] (Iverson, 2000); [2](Goren and Aharonov, 2007); [3](Handwerger et al., 2013); [4](Berti and Simoni, 2012); [5](Reid, 1994); [6](Kim et al., 2010); [7](Hu et al., 2019); [8](Schulz et al., 2009); [9](Baum et al., 2010); [10](Finnegan et al., 2021); [11](Chien-Yuan et al., 2005); [12](Jiménez-Martínez et al., 2013); [13](Pacheco, 2013). Hillslope length in Taiwan is measured by considering the nearest hydrological distance between crests and rivers, considering that the transition between rivers and hillslope occurs at 0.9 km².**

## 5.7 The case of typhoon Morakot

As already mentioned, typhoon Morakot triggered more than 10,000 landslides in the south of Taiwan, leading to major damage and casualties (Lin et al., 2011; Hung et al., 2018; Yang et al., 2018). Landslides triggered by this event show a wide range of size, spanning from 576 $m^2$ to almost 2.5 km². with a PDF peaking around 1000 $m^2$. Most of the failures occurred on slopes between 30° to 40° (Lin et al., 2011). One of the biggest landslides reached depths above 86 m, buried the Shiaolin village and caused around 400 deaths (Tsou et al., 2011; Lin and Lin, 2015). Many studies point out the role of the exceptional accumulation of rainfall during the typhoon, up to 3 m in the south of the island (Tsou et al., 2011; Yang et al., 2018). However, the hydrogeological context in which the event occurred is often overlooked. Indeed, Morakot followed a relatively dry period, with no recorded precipitations over the Shiaolin village during the two months prior the typhoon (Kuo et al., 2013). This had

an impact on the water tables along hillslopes, which were likely at a low level from our modelling results and allowed for high pore pressure changes $\psi_{rain}$, where hillslopes would otherwise have already been saturated in a wet context.

The devastating effect of typhoon Morakot might be due to the combination of large precipitations and a deep water table accommodating large pore pressure variations under hillslopes.

**6 Conclusion**

We developed a model to assess the respective role of hydrological and atmospheric forcing on slope stability. This model, based on a 2D hydrological computation of the water table, is an improvement to the well-known 1D infinite slope, for it allows to better account for the along-slope geometry of the water table, and its temporal variations following typhoons. We then used 1D diffusion equations to simulate pore pressure variations induced by rainfall and atmospheric perturbations.

The model was applied to several typhoons that struck Taiwan to understand the failure mechanisms leading to landsliding. Consistent with previous studies (Vassallo et al., 2015), results show that rainfall can lead to large pore pressure increases – more than 100 kPa in the case of typhoon Krosa – especially towards the crest of the slope, where the water table elevation gains are maximum. On the other hand, for similar typhoons, atmospheric induced pore pressure is usually around 1 kPa all along the slope, 1 to 2 orders of magnitude less than the rainfall contribution. However, the rainfall history plays a key role

when assessing slope stability. Indeed, many typhoons strike over already fully saturated slopes, especially during or after the wet season, preventing further infiltration and leaving the atmospheric-induced pore pressure as the main destabilizing factor. In broader terms, if models show that saturated slopes with low diffusivity could potentially fail simply in response to atmospheric pressure drop, rainfall infiltration remain by far the dominant destabilizing factor for relatively dry slopes with high diffusivity. As a striking example, our results show that typhoon Morakot occurred after a relatively dry period, leading

to significant infiltration, water table rise and pore pressure increase, especially towards the toe of the slopes. Accounting for such a groundwater dynamic is fundamental to explain the large number of triggered landslides that ruptured close to the hillslope toes (West et al., 2011). Our model outcomes also corroborate the preferential location of storm-triggered landslides at the toe of hillslopes (Meunier et al., 2008). As a long-term insight, we believe that a better characterisation of the timing of landslide failure during large storms or typhoons, for instance thanks to the development of SAR imagery (Singhroy and

Molch, 2004; Xu et al., 2019; Esposito et al., 2020), could help to separate the respective role of atmospheric pressure drop and rainfall in slope destabilisation.

## Appendix A

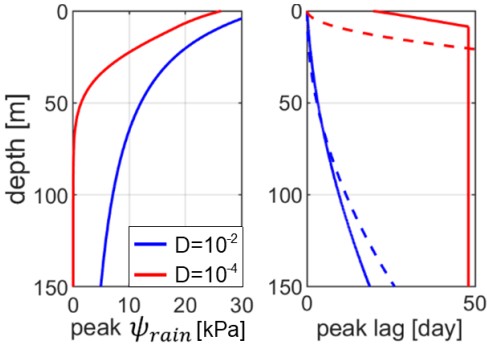

**Figure A1: Maximum rainfall response in function of depth and its time lag for the synthetic recharge (Fig. 2). The time lag caps at 48 days, maximum duration between the end of the recharge (2 days) and the length of the time vector (50 days). Dashed lines represent theorical characteristic response times $t_c = z^2/D$, in comparison to the times of maximum response computed from the model.**

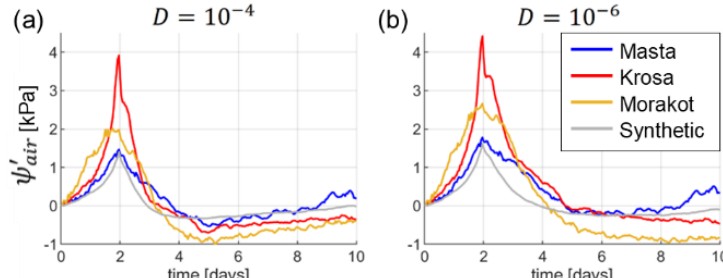

**Figure A2: Atmospheric induced pore pressures $\psi'_{air}$ for each typhoon event in the finite hillslope model, 5m under the topography (a) at a diffusivity of $10^{-4}\ m^2\ s^{-1}$, and (b) at a diffusivity of $10^{-6}\ m^2\ s^{-1}$. At these diffusivities, the hillslope is fully saturated from its initial state. Therefore, no rainfall effects are associated with the events, and $\psi'_{air}$ is the same at the toe and the crest of the hillslope.**

## Competing interests

The authors declare that they have no conflict of interest.

## Code and data Availability

The code and the data used for this study are available at https://doi.org/10.5281/zenodo.5654768

## Author contribution

LP and PS conceptualized the mechanical stability model. LP and LL worked on the hydrological model. MM helped by providing data. The codes were written by LP. All authors participated in improving the paper by editing.

## Acknowledgements

This project has received funding from the European Research Council (ERC) under the European Union's Horizon 2020 research and innovation program (grant agreement no. 803721). The authors acknowledge the Taiwan Typhoon and Flood Research Institute, National Applied Research Laboratories, for providing the Data Bank for Atmospheric & Hydrologic Research service (https://www.narlabs.org.tw/). We also acknowledge support by the France-Taiwan International Associate Laboratory "From Deep Earth to Extreme Events".

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
