# Peer review of "Finite-hillslope analysis of landslides triggered by excess pore water pressure: the roles of atmospheric pressure and rainfall infiltration during typhoons"

_Natural Hazards and Earth System Sciences, 2021_

## Referee Comment (RC1)

Review of "Modelling the control of groundwater on landslides triggering: the respective role of atmosphere and rainfall during typhoons"

Summary:

In this contribution, Pelascini et al. examined the effects of pore pressure changes due to atmospheric pressure and rainfall infiltration on the stability of hillslopes of finite length. Here the Mohr-Coloumb failure criterion was the conceptual basis of stability, though throughout the paper, emphasis was placed on pore pressure components of the effective stress rather than the failure criterion. Time-evolution of pore pressure at the hillslope crest and toe were calculated by convolving analytical solutions for groundwater flow and diffusion of pore pressure with synthetic and real timeseries of rainfall and atmospheric pressure. The results showed that the importance of the two dynamic pore pressure-generating mechanisms (atmospheric pressure and infiltration) varied in space on the slope, largely driven by differences in depth to the water table, and in time depending on the mechanism's response timescale. The results suggest that more attention should be paid to slope stability effects of atmospheric pressure fluctuations in large storms, and that estimates of landslide timing in relation to pressure fluctuations and precipitation could help distinguish drivers of landslides.

My experience makes me most suited to comment on the groundwater hydrology aspects of this paper, rather than the landslide hazard component. In this respect I have a few concerns.

1. The hydrological model used in this paper is a combination of a Dupuit-Forchheimer (D-F) aquifer model and a one-dimensional infiltration model based on the Richards equation. After reading the paper I was left unclear on how exactly these two models interact, and what effects the transient component of the water table response has on their results.
2. There needs to be more careful attention paid to the relationship between this hydrological model and the expected groundwater dynamics of the landscapes the model intends to capture. (Steep landscape hillslope hydrology see e.g. Montgomery et al. (1997)). The linearized, horizontal-based form of the D-F model may be appropriate in low relief settings or for deep aquifers that respond slowly to recharge, but the landscapes considered here are steep, and recharge here is assumed to infiltrate instantaneously to the water table.
3. The two hydrological models operating together contain potentially contradictory information on the pore water pressure below the water table.

While I recognize that issues 2 and 3 are acknowledged in Discussion section 5.1, it seems that most of the paper does not meaningfully engage with these limitations. If the authors retain the current hydrological model, rationale and limitations of the model need to be more clearly stated up front in the introduction and methods sections. While I cannot comment on the novelty of the landslide hazards component of this paper, I was left with the impression that there is merit to exploring the processes they consider here, though I think the hydrological basis of this work could use more thought. I've added more details in the line-by-line notes below.

Line by line:

*Title:*
Title feels a little unspecific – what about the atmosphere, and what about groundwater? Could I suggest something along the lines of: "Finite-hillslope analysis of landslides triggered by excess pore water pressure: the roles of atmospheric pressure and rainfall infiltration during typhoons"

*Abstract:*

Two things in the abstract seem contradictory to me. Please reconcile or clarify the following:

Lines 10-11 you state that "atmospheric pressure changes and rainfall induced groundwater level change can generate pore pressure changes with similar amplitude," but then in line 17-18, you say they differ by perhaps several orders of magnitude.

Lines 14-15 you state that "rainfall infiltration and atmospheric pressure variations" are described by diffusion equations, but then in line 18 you say the effects of atmospheric pressure are instantaneous. This may be a matter of the phrasing, but it is confusing.

*Introduction:*

Line 31: "cumulated rainfall" -> "groundwater recharge"

Line 38: "Little attention has been by received by this potential slope destabilisation factor" -> "This slope destabilisation factor has received little attention."

Line 40: "…modifying slope stability." Citation?

Line 55-56: "As both rainfall and atmospheric effects implies pore pressure diffusion in groundwater, the link to slope stability requires a specific model." This sentence seems vague to me.

Line 59: "allows *us* to define"

Line 62: remove "about"

*Methods:*

Line 65: Not sure what "homogenous half space" means and it is not mentioned anywhere else in the text.

Line 90: "Under rainfall constrain" ?

Line 101: "hydrogeological model" usually refers to a model of the characteristics of an aquifer – it's permeability, porosity, stratigraphy and composition (e.g., Condon et al. 2021 5.1). Maybe hydrological model would be better?

Line 101: I would use "slope" or "topographic slope" over "dip," because dip has a different geologic meaning.

Line 104: Interesting, I have not seen this called the diffusivity equation before. Looking around online, it seems this term is more commonly applied in the petroleum industry to other fluids? In hydrology I see this called the Boussinesq equation (e.g. Troch 2013, paragraph 9, Boussinesq equation for horizontal aquifers) or simply the Dupuit-Forchheimer equation.

Line 110: "storage" -> "storage coefficient"

Line 112: "in term of" -> "in terms of"

Line 119, 130: "in function of" -> "as a function of"

Line 131: It's unclear to me whether you use this solution, given the discussion in 2.3, where it seems that only $h_s$ matters. Does the static pressure head in response to recharge come into effect in your model?

Line 135: This sounds you are disregarding the transient component of the water table variation in the Dupuit-Forchheimer model? Or are you only disregarding its affect on pressure head and not on water table position?

Lines 137-138: It is not necessarily described by diffusion. In Iverson (2000) there are extensive assumptions and conditions required to reduce the Richards equation to this particular 1D diffusion form. These need to be identified and discussed.

Line 138: "characterise" -> "characterised"

Line 139: "model considered a 2D mode" Model? Consider rephrasing to avoid repetition.

Line 141: "one-dimension" -> "one-dimensional"

Lines 148-149: Could you more clearly state the boundary conditions to arrive at this solution? The constant loading gives the surface boundary condition, what is the condition at depth? Seems like this solution is not accounting for the water table depth?
Line 152: tc = z2/D should this be \hat{D}?
Line 154: "convoluted" -> "convolved"
Line 167: Again more clearly state lower boundary condition.
Line 168: change citation type to "Carslaw and Jaeger (1959)"

Figure 1:
Can you label hydraulic head *h*?

*Results – Synthetic:*
Line 176: "toe of and at the very top of" I would call the top either *crest* or *ridge.*
Line 177: I think some more elaboration of this consideration of the slope at yield is needed. It seems critical to how your are interpreting the results.
Line 183: Don't need the figure description in parenthesis.
Line 185: "an 86.4 mm cumulated rainfall" -> "86.4 mm of accumulated rainfall"
Line 197: Underestimation of tc... Can you provide more insight into the physical meaning of tc here? Semantically, it also seems to me less that tc is underestimated, and more that it may not be the right quantity for comparison with the timescale estimated.
Line 200: "slop" -> "slope"
Line 214: Still unclear exactly how the water table rise during event is incorporated into your model.

*Results – Application:*
Line 247: "east of Taiwan" -> "eastern Taiwan" or "the east of Taiwan"
Line 251: "inferior to" -> less than
Line 255: "contrasted" -> contrasting
Line 257: remove "has"

Figure 5: Great plot on the left – I like how your selected storms are a kind of envelope around the extreme events.
Section 4.2:
- How did you select hillslope length? How sensitive are results to hillslope length? Can you use your analytical solutions to show something about this?
- Is there evidence in the literature or in published well/piezometer data that hillslopes fully saturate during typhoons?
- Provide the equation for the infinite slope model used for comparison
- Can you say anything from your model about which storms cause landslides and which ones don't?

Line 278: you say "amount of rainfall" which to me implies rainfall depths, but rates are given. I would make these agree.
Line 290: "caps off" colloquial language, consider replacing
Line 296: "in function of the event" what does this mean?

Figure 6: The use of black and blue together in plots b-i is difficult to read. I would choose a better color contrast.

*Discussion:*

Line 314: "models limitations" -> "model limitations"

Line 315: "considered in this study consider" rephrase

Line 355: "has been" -> was

Line 365: Worth mentioning in this section that the diffusivity in the 1D model is Iverson (2000)'s maximum hydraulic diffusivity, derived for conditions near saturation.

Line 385: Can you provide some more physical insight here on why diffusivity affects these in opposite ways?

Line 392: When considering only these two effects. Do you think atmospheric pressure effects could be more important than other mechanisms going on when hillslopes fully saturate, like seepage? (Found this, line 455-456)

Lines 399: When you say that the response of psi_air is instantaneous, I think that then the pore pressure response to a gate function should just look like the gate function. But it seems like what you're implying is that there is no delay in the beginning of the response, even though there still is a decay of the response in time?

Lines 406-408: How does this finding compare with literature? Do we see landslides occurring in these locations?

Line 431: "dominants" -> dominant

Line 443, 449: "repartition" Partitioning? Check word choice.

Line 450: "typhon induced" -> "typhoon-induced"

Line 478: weeks or months after the rain event – has this been observed? I think a reference would strengthen this argument.

Line 489: "amount of cumulated rainfall" -> "depth of rainfall" or "accumulation of rainfall"

Line 495: large variations in pore water pressure?

Works Cited:

Condon, L. E., Kollet, S., Bierkens, M. F. P., Fogg, G. E., Maxwell, R. M., Hill, M. C., et al. (2021). Global Groundwater Modeling and Monitoring: Opportunities and Challenges. *Water Resources Research*, *57*(12), e2020WR029500. https://doi.org/10.1029/2020WR029500

Iverson, R. M. (2000). Landslide triggering by rain infiltration. *Water Resources Research*, *36*(7), 1897–1910. https://doi.org/10.1029/2000WR900090

Montgomery, D. R., Dietrich, W. E., Torres, R., Anderson, S. P., Heffner, J. T., & Loague, K. (1997). Hydrologic response of a steep, unchanneled valley to natural and applied rainfall. *Water Resources Research*, *33*(1), 91–109. https://doi.org/10.1029/96WR02985

Troch, P. A., Berne, A., Bogaart, P., Harman, C., Hilberts, A. G. J., Lyon, S. W., et al. (2013). The importance of hydraulic groundwater theory in catchment hydrology: The legacy of Wilfried Brutsaert and Jean-Yves Parlange. Water Resources Research, 49(9), 5099–5116. https://doi.org/10.1002/wrcr.20407

---

## Referee Comment (RC2)

(8b)

[referee-annotated manuscript omitted]

---

## Author Response (AR1)

**Author's response to referees**

**Response to Referee 1**

Summary:
In this contribution, Pelascini et al. examined the effects of pore pressure changes due to atmospheric pressure and rainfall infiltration on the stability of hillslopes of finite length. Here the Mohr-Coloumb failure criterion was the conceptual basis of stability, though throughout the paper, emphasis was placed on pore pressure components of the effective stress rather than the failure criterion. Time-evolution of pore pressure at the hillslope crest and toe were calculated by convolving analytical solutions for groundwater flow and diffusion of pore pressure with synthetic and real timeseries of rainfall and atmospheric pressure. The results showed that the importance of the two dynamic pore pressure-generating mechanisms (atmospheric pressure and infiltration) varied in space on the slope, largely driven by differences in depth to the water table, and in time depending on the mechanism's response timescale. The results suggest that more attention should be paid to slope stability effects of atmospheric pressure fluctuations in large storms, and that estimates of landslide timing in relation to pressure fluctuations and precipitation could help distinguish drivers of landslides.

**We are grateful to the referee for this detailed and precise comments on the preprint.**

My experience makes me most suited to comment on the groundwater hydrology aspects of this paper, rather than the landslide hazard component. In this respect I have a few concerns.
1. The hydrological model used in this paper is a combination of a Dupuit-Forchheimer (D-F) aquifer model and a one-dimensional infiltration model based on the Richards equation. After reading the paper I was left unclear on how exactly these two models interact, and what effects the transient component of the water table response has on their results.
2. There needs to be more careful attention paid to the relationship between this hydrological model and the expected groundwater dynamics of the landscapes the model intends to capture.(Steep landscape hillslope hydrology see e.g. Montgomery et al. (1997)). The linearized, horizontal-based form of the D-F model may be appropriate in low relief settings or for deep aquifers that respond slowly to recharge, but the landscapes considered here are steep, and recharge here is assumed to infiltrate instantaneously to the water table.
3. The two hydrological models operating together contain potentially contradictory information on the pore water pressure below the water table.

While I recognize that issues 2 and 3 are acknowledged in Discussion section 5.1, it seems that most of the paper does not meaningfully engage with these limitations. If the authors retain the current hydrological model, rationale and limitations of the model need to be more clearly stated up front in the introduction and methods sections. While I cannot comment on the novelty of the landslide hazards component of this paper, I was left with the impression that there is merit to exploring the processes they consider here, though I think the hydrological basis of this work could use more thought. I've added more details in the line-by-line notes below.

**Our goal was to better understand the specific (physical) controls of rainfall and atmospheric pressure changes on landslide triggering.**

**In geomorphology, the 1D diffusion model (Iverson model) remains as a reference for rainfall-driven landslides. Though, this model has strong limitations (infinite slope, i.e. local point of view, inability to consider both recharge and GW upflow, define initial conditions …) and therefore difficult to use at hillslope scale.**

Our point of view was to develop a simple model, as simple as the original Iverson model, which could better represent groundwater dynamics under hydrological and atmospheric boundary conditions. As a consequence, we are far from the quality of a "site model", as we rather define physical controls. Following this work, we are now setting up a modflow-based model on a Taiwanese catchment, driven by a land-surface model, to better represent actual groundwater flow, and also interception with the surface.

It appears the goal of the paper is not stated clearly enough, and this causes confusion for the reader. Some clarifications are indeed needed and will be added in the first parts of the manuscript. The goal is here to consider the water table variations through a simple analytical model in order to improve slope stability assessments.

1. The Dupuit-Forchheimer (DF) model considers horizontal flows and defines the water table surface. The 1D diffusion models allows for the computation of pressure propagation through the groundwater. The two models do not interact, the rainfall pressure diffusion feeds on the output of the DF model.
   ⇨ Additional explanations are provided lines 88-91, 150-152, 175-177 and figure 1 has been modified to further illustrate how the models are implemented.

2. The hydrological model here is indeed basic. It is based on the hypothesis of negligible vertical flow, which might be suitable for thick aquifers and/or low angle slopes. The hypothesis might not be valid in Taiwan. While this doesn't represent the complexity of the geology for the slopes in Taiwan, it has been deemed necessary to keep the model simple.
   ⇨ We clarify this when we introduce the hydrogeological model (lines 121-126)

3. The two pore pressure diffusion models indeed predict different pore pressures in response to the different forcing, but the solution of the diffusivity equation can be added by linearity. Physically this can lead to a pressure gradient and groundwater flows in two opposite directions.
   ⇨ This limitation of the model is addressed in the discussion (lines 351-354).

Line by line:
Title:
Title feels a little unspecific – what about the atmosphere, and what about groundwater? Could I suggest something along the lines of: "Finite-hillslope analysis of landslides triggered by excess pore water pressure: the roles of atmospheric pressure and rainfall infiltration during typhoons"

We agree with this comment and have changed the title accordingly.

Abstract:
Two things in the abstract seem contradictory to me. Please reconcile or clarify the following:
Lines 10-11 you state that "atmospheric pressure changes and rainfall induced groundwater level change can generate pore pressure changes with similar amplitude," but then in line 17-18, you say they differ by perhaps several orders of magnitude.

Thank you for pointing this out. When thinking about pressure change, 10 hPa atmospheric pressure change is equivalent to a 100-mm rainfall event. Therefore, atmospheric effects can generate similar pore pressure amplitudes as rainfall, in some specific cases. However, the rainfall effects generally reach values of pore pressure orders of magnitude above the atmospheric ones because the latter is function of the derivative of the atmospheric pressure, and atmospheric

**pressure rarely drops in an instant. The sentence line 12 has been modified to prevent this potential confusion.**

Lines 14-15 you state that "rainfall infiltration and atmospheric pressure variations" are described by diffusion equations, but then in line 18 you say the effects of atmospheric pressure are instantaneous. This may be a matter of the phrasing, but it is confusing.

**The phrasing can indeed be confusing. Both rainfall and atmospheric effects are described with diffusion equations. However, the atmospheric effect is instantaneous due to the transfer of the pressure load from the atmosphere to the pores through the skeleton or solid phase of the medium. The decay of this change in pore pressure is described with diffusion equations. This is now clarified lines 18-20.**

**Introduction:**
Line 31: "cumulated rainfall" -> "groundwater recharge"
**Done**
Line 38: "Little attention has been by received by this potential slope destabilisation factor" -> "This slope destabilisation factor has received little attention."
**Done**
Line 40: "…modifying slope stability." Citation?
**The work that was been referred to here was Schultz et al. (2009), and the citation has been added.**
Line 55-56: "As both rainfall and atmospheric effects implies pore pressure diffusion in groundwater, the link to slope stability requires a specific model." This sentence seems vague to me.
**The sentence has been clarified (lines 57-58).**
Line 59: "allows us to define"
**Done**
Line 62: remove "about"
**Done**

**Methods:**
Line 65: Not sure what "homogenous half space" means and it is not mentioned anywhere else in the text.
**"Infinite homogeneous slope" might be more self-explanatory indeed.**
Line 90: "Under rainfall constrain" ?
**"under rainfall forcing"**
Line 101: "hydrogeological model" usually refers to a model of the characteristics of an aquifer – it's permeability, porosity, stratigraphy and composition (e.g., Condon et al. 2021 5.1). Maybe hydrological model would be better?
**We agree, "hydrological model" is more appropriate since no assumption on the structure and stratigraphy are made.**
Line 101: I would use "slope" or "topographic slope" over "dip," because dip has a different geologic meaning.
**Done**
Line 104: Interesting, I have not seen this called the diffusivity equation before. Looking around online, it seems this term is more commonly applied in the petroleum industry to other fluids? In hydrology I see this called the Boussinesq equation (e.g. Troch 2013, paragraph 9, Boussinesq equation for horizontal aquifers) or simply the Dupuit-Forchheimer equation.
**We used the term diffusivity equation by default since it describes this physical phenomenon, but indeed, it is generally named after Boussinesq in hydrology. It will therefore be referred to as Boussinesq's equation in order to fit the hydrology community convention.**
Line 110: "storage" -> "storage coefficient"

**Done**

Line 112: "in term of" -> "in terms of"

**Done**

Line 119, 130: "in function of" -> "as a function of"

**Done**

Line 131: It's unclear to me whether you use this solution, given the discussion in 2.3, where it seems that only h_s matters. Does the static pressure head in response to recharge come into effect in your model?

**This comment corresponds to the same issues raised in the general comment #1. The models needed to be presented with more clarity. The static position h_s of the water table is taken into account to define the water table initial position. The water table level is then updated using the transient solution h_t. The static pressure head (due to h_s) does not create dynamic pore pressure, and is not investigated in this study. Indeed, the focus of this work is the dynamic transient effects – this is why the slope is considered at yield – an no absolute values of safety factor are computed. We deliberately only modelled the transient/dynamic effects due to changes in water table level h_t inducing new loadings. These loadings are then fed into the diffusion model to propagate in depth.**

⇨ **Precisions have been added lines 150-152, and the use of transient water table into the rainfall-induced pore pressure diffusion model is explained lines 175-177 with the help of a new graph in Fig. 1.**

Line 135: This sounds you are disregarding the transient component of the water table variation in the Dupuit-Forchheimer model? Or are you only disregarding its affect on pressure head and not on water table position?

**This is answered in the previous comment.**

Lines 137-138: It is not necessarily described by diffusion. In Iverson (2000) there are extensive assumptions and conditions required to reduce the Richards equation to this particular 1D diffusion form. These need to be identified and discussed.

**Indeed, the propagation of pore pressure can be described by pressure diffusion – which is what Iverson proposed in his model, under the main hypothesis of an infinite slope geometry, and wet initial conditions, so that there are no changes in hydraulic conductivities above and below the water table. Then flow is described using Richard's equation and reduced to a one-dimensional diffusion equation.**

⇨ **These hypothesis are now mentioned line 157.**

Line 138: "characterise" -> "characterised"

**Done**

Line 139: "model considered a 2D mode" Model? Consider rephrasing to avoid repetition.

**Done**

Line 141: "one-dimension" -> "one-dimensional"

**Done**

Lines 148-149: Could you more clearly state the boundary conditions to arrive at this solution? The constant loading gives the surface boundary condition, what is the condition at depth? Seems like this solution is not accounting for the water table depth?

**The upper boundary condition at the surface is a Neumann condition (known flow). There is no lower boundary condition, the solution used here is for the diffusion through a semi-infinite solid (therefore no lower boundary), as the aquifer is taken with a sufficient thickness for the horizontal DF model and its lower boundary can be considered at infinity with regards to the depths investigated here.**

⇨ **We now clearly state the boundary conditions of the model lines 165-166**

Line 152: tc = z2/D should this be \hat{D}?

**The characteristic time, as presented in several studies (Iverson, 2000; Handwerger et al., 2013), is indeed tc = z2/D . Its definition has been rewritten in a clearer manner lines 161-163.**

Line 154: "convoluted" -> "convolved"

**Done**
Line 167: Again more clearly state lower boundary condition.
**As for the commentary relative to lines 148-149, there is no lower boundary condition, it is considered to infinity. The upper boundary this time is a Dirichlet condition. As for the previous commentary, additional explanation as well as more precise citation are provided lines 192-193.**
Line 168: change citation type to "Carslaw and Jaeger (1959)"
**Done**
Figure 1:
Can you label hydraulic head h?
**Done**

**Results – Synthetic:**
Line 176: "toe of and at the very top of" I would call the top either crest or ridge.
**Required changes have been done, the top of the slope is now been referred as crest in the manuscript.**
Line 177: I think some more elaboration of this consideration of the slope at yield is needed. It seems critical to how your are interpreting the results.
**Yes, this is a critical part in this study. The slope is considered at yield in order to investigate only dynamic variations and not static effects (as already discussed in response to the commentary on line 131). This is done so that the results of the models are not dependent on the intrinsic mechanic and topographic properties of the slope.**
⇨ **We have clarify that in the manuscript. A discussion about the reason why we consider a slope at yield has also been added in the description of the failure mechanism lines 92-94.**
Line 183: Don't need the figure description in parenthesis.
**Indeed, this was a mistake and should not have appear here.**
Line 185: "an 86.4 mm cumulated rainfall" -> "86.4 mm of accumulated rainfall"
**Done**
Line 197: Underestimation of tc… Can you provide more insight into the physical meaning of tc here? Semantically, it also seems to me less that tc is underestimated, and more that it may not be the right quantity for comparison with the timescale estimated.
**We fully agree with this remark. The characteristic time tc is not really appropriate to these comparisons. It represents the minimum time at which a strong pore pressure occurs at the depth z, which is why it is compared to the maximum response time. However, Handwerger et al. (2013) showed it corresponds to "the time 48% of the surface forcing is felt at a given depth". It still gives an idea of the timing of the diffusion. The definition and a discussion about tc is provided lines 221-224.**
Line 200: "slop" -> "slope"
**Done.**
Line 214: Still unclear exactly how the water table rise during event is incorporated into your model.
**This is the same issue as addressed in the general comment #1 and the comments concerning lines 131 and 135. The description of the models is clarified.**

**Results – Application:**
Line 247: "east of Taiwan" -> "eastern Taiwan" or "the east of Taiwan"
**Done**
Line 251: "inferior to" -> less than
**Done**
Line 255: "contrasted" -> contrasting
**Done**
Line 257: remove "has"
**Done**
Figure 5: Great plot on the left – I like how your selected storms are a kind of envelope around the

extreme events.

Section 4.2:
- How did you select hillslope length? How sensitive are results to hillslope length? Can you use your analytical solutions to show something about this?
**The hillslope length L, in the equations presented by Townley (1995) does not have an influence on the form of the static water table. Only the position along the hillslope – namely the parameter x/L – impacts the water table response. The transient water table variation would keep the same form but see its response time changing accordingly to the diffusion characteristic time over the full hillslope length $L^2/D$. The hillslope length has been set as 500 m as it seems to represent hillslopes in Taiwan quite well (Fig. 8 (a)). We now discuss the scale of the hillslope lines 525-529.**
- Is there evidence in the literature or in published well/piezometer data that hillslopes fully saturate during typhoons?
**To our knowledge, no study or data monitoring water table of hillslopes undergoing typhoons with high spatial and temporal resolution exist. But this is a very important question and we are exploring whether sentinel satellites can define such behaviour.**

- Provide the equation for the infinite slope model used for comparison
**The equation of diffusion of pore pressure for the infinite slope model is the same as the one used in the finite hillslope model (equations 7 and 8), as specified in the manuscript. It does not seem necessary to provide the equation a second time.**
- Can you say anything from your model about which storms cause landslides and which ones don't?
**The model shows the greatest rainfall-induced pore pressure are achieved when weather events strike hillslopes with a deep water table level, which allow for large water table variations. The atmospheric effect is more pronounced when the drop of atmospheric pressure is rapid. Therefore, according to the model, a sudden and violent storm occurring after a drought period is the most prone to trigger a landslide. On the other hand, a storm striking a fully saturated hillslope with a gradual drop of atmospheric pressure is less likely to cause slope failure. However, this model does not take the static components of the safety factor into account, nor the seepage forces, and these might cause a fully saturated hillslope to fail.**
Line 278: you say "amount of rainfall" which to me implies rainfall depths, but rates are given. I would make these agree.
**Values given are indeed rates. They correspond to the mean rate during the past 6 months before the typhoons. The sentence has been corrected.**
Line 290: "caps off" colloquial language, consider replacing
**Done**
Line 296: "in function of the event" what does this mean?
**We have rephrased as: "depending on the event"**
Figure 6: The use of black and blue together in plots b-i is difficult to read. I would choose a better color contrast.
**Done**

Discussion:
Line 314: "models limitations" -> "model limitations"
**Done**
Line 315: "considered in this study consider" rephrase
**Done**
Line 355: "has been" -> was
**Done**
Line 365: Worth mentioning in this section that the diffusivity in the 1D model is Iverson (2000)'s maximum hydraulic diffusivity, derived for conditions near saturation.

**Hydraulic diffusivity in Iverson's model is indeed the maximum hydraulic diffusivity, and is the diffusivity considered along this paper. This is added in the description of the rainfall-induced diffusion model (Sec. 2.3), line 160.**

Line 385: Can you provide some more physical insight here on why diffusivity affects these in opposite ways?

**The impact on diffusivity on each effect is developed and explained just above, and it doesn't seem to require any modifications in that way.**

Line 392: When considering only these two effects. Do you think atmospheric pressure effects could be more important than other mechanisms going on when hillslopes fully saturate, like seepage? (Found this, line 455-456)

**This is a very good question also raised by the other reviewer. The intensity of seepage has not been modelled in this study, which is focused on the two effects of Psi_rain and Psi_air'. However, the atmospheric effects will never exceed the atmospheric pressure change, which is unlikely to change more than 5 kPa. The seepage is function of the flow of groundwater, and might surpass these values.**

Lines 399: When you say that the response of psi_air is instantaneous, I think that then the pore pressure response to a gate function should just look like the gate function. But it seems like what you're implying is that there is no delay in the beginning of the response, even though there still is a decay of the response in time?

**Yes, the response to atmospheric effects is instantaneous because of the load transfer through the skeleton, but fades slowly by diffusion. The response to the gate function at a low diffusivity (D=10e-6 m²/s) is almost identical to the gate function, because little diffusion occurred during the forcing.**

Lines 406-408: How does this finding compare with literature? Do we see landslides occurring in these locations?

**This statement was referring to the model's outcome, not to actual data. No studies were found reporting landslides locations under these conditions.**

Line 431: "dominants" -> dominant

**Done**

Line 443, 449: "repartition" Partitioning? Check word choice.

**The word has been changed to "distribution".**

Line 450: "typhon induced" -> "typhoon-induced"

**Done**

Line 478: weeks or months after the rain event – has this been observed? I think a reference would strengthen this argument.

**This has been reported in the context of long-term forcing in slow-moving landslides (Iverson and Major, 1987), but not observed for typhoons.**

Line 489: "amount of cumulated rainfall" -> "depth of rainfall" or "accumulation of rainfall"

**Done**

Line 495: large variations in pore water pressure?

**Yes, this was referring to the possibility of the water table in the case of Morakot to accommodate for large variations in water table level, and therefore in pore pressure. The sentence has been corrected (line 547).**

The subject manuscript describes a modeling study focused on identifying relative (and absolute) contributions of rainfall and atmospheric pressure change during typhoons in causing landslides. Consideration of atmospheric pressure effects on slope stability is quite novel; I'm aware of only one other study related to this subject (cited in paper). During typhoons, atmospheric pressure can drop several kPa, causing reduced slope stability by reducing effective normal stress, whereas rainfall amounts may contribute several tens (or more) of kPa to reduce this normal stress. However, by using simple 2D and 1D hydrogeologic modeling, the authors show that these effects on effective normal stress vary through time and slope position, such that differing initial water table conditions and hydraulic diffusivity of hillslope materials may result in relatively greater or lesser effects on slope instability from rainfall and atmospheric pressure change. The manuscript therefore presents important new insights into landslide triggering factors. Unfortunately, the model is poorly tested by empirical evidence, primarily because of the lack of data available to identify landslide timing and potential triggering during typhoons when rainfall and atmospheric pressure drop both are significant.

1. **We thank the reviewer for his comments. This specific comment summarizes well the novelty and limits of our manuscript. The model we use has indeed only be tested against data from a slow-moving landslide (Schulz et al., 2009) for small atmospheric pressure changes. Our manuscript uses this tested model in the case of larger atmospheric pressure changes (i.e., typhoons) and consider mostly the case of catastrophic landslides. For these landslides, there is to our knowledge (as also stated by the reviewer) no clear evidence of landslides triggered by atmospheric pressure changes, potentially due to the current inability of observations to resolve the timing of landslides during large storms.**
   **The goal of the modelling approach was (1) define a simple model, as simple as the original Iverson model, which could better represent groundwater dynamics under hydrological and atmospheric boundary conditions (2) define the main physical controls. This allows to underline observation requirements which could better inform on actual processes in landslide generation.**
   ⇨ **We now clearly state that the model has not been tested against natural catastrophic landslides (Lines 182-184)**

The conceptual and mathematical models developed in the manuscript involves deep groundwater within a homogeneous hill (peak to valley), discounting oftentimes perched groundwater within regolith, where many landslides generate during/following intense rainfall. How representative of the test locations in Taiwan is this conceptual model? What implications/omissions exist with respect to the lack of consideration of regolith?

2. **This is a very good point. Our model considers a homogeneous hillslope with no lithology change above a single unconfined aquifer. This hypothesis comes from the hydrological model, which considers an aquifer relatively thick in comparison to the 500 m length of the hillslope so that the groundwater flow can be described as horizontal.**
   **This simplification indeed does not fully account for the hydrogeological complexity under Taiwan's hillslopes but is necessary to develop a model rooted on basic analytical solutions.**
   **However, the model can be applied at any scale and therefore represent smaller aquifer – including perched ones – at smaller scales, as long as the boundary conditions and the hypothesis of the hydrological model are respected. Considering smaller aquifers, for instance developed in the regolith, and depending on their connexion with the hillslope**

**aquifer, this would lead to similar water table profile yet with shorter characteristic times, following this scaling $L^2/D$, where L is the length of the considered perched aquifer. Following this work, we are now setting up a modflow-based model on a Taiwanese catchment, driven by a land-surface model, to better represent actual groundwater flow in heterogeneous systems.**

⇨ **We now acknowledge this simplification and limitation and discuss the scale effect in the discussion (lines 347-350 & 525-529) and illustrated it with a new figure (Fig. 8)**

The 2D and 1D models differently treat application of transient atmospheric pressure and rainfall, with diffusion occurring from different locations (ground surface or water table) depending on the forcing. Section 5.2 summarizes some of these differences and indicates that model uncertainty might explain some field observations. The dramatic differences between the models and, sometimes, their output makes me wonder what results may be believed, as well as wonder why the modeling approaches were not more consistent. A more critical evaluation of the implications of their differences is warranted with respect to both magnitudes and timing of effective normal stress change. For example, lines 349-350 indicate that the lack of an infiltration model and application of rainfall immediately and entirely at the water table "might underestimate the response time." Actually, except in special circumstances, these factors definitely underestimate the response time and by variable amounts ranging likely over several orders of magnitude for realistic conditions. What are the overall impacts of the simplifications involved with the modeling?

3. **We acknowledge this comment. The 2D hillslope model mainly differs from the infinite slope model by having a dynamic and fluctuating water table, and especially to represent the impact of uphill groundwater flow contributing to local pressure change, in addition to local recharge. This is very important, and translates as critical when considering the specific behavior at the toe and crest of the hillslope. This leads to a key difference as the 2D model uses the water table variations to feed into the pore pressure diffusion. The underlying hypothesis here is that pore pressure variations induced by rainfall are caused by changes in the height of the water table. As a consequence, since the pore pressure is applied from the water table, the infiltration time needed for rainfall to reach the water table is disregarded. Indeed, this underestimates the response time for the areas where the water table is deep – namely the upper part of the hillslope – and does not allow to describe shallow landslides.**

⇨ **We now clearly discuss this limitation of the 2D model in the discussion comparing 1D and 2D models (lines 370-375)**

One or two lines mention that seepage forces may be important contributors to slope instability, and they certainly are, especially near discharge zones near slope toes (e.g., Iverson 2000, cited in paper). The importance of non-hydrostatic gradients in slope instability should be emphasized, especially with respect to landslide triggering from regions near slope toes. Additionally, please see next comment regarding landslides near slope toes.

4. **Indeed, the seepage effect leads to slope instabilities and is discussed in many papers. Loss of stability by seepage is mainly related to vertical flow, creating a force against the weight of the slope; yet the hydrological model only considers horizontal flows. Approximation of seepage forces based on such a model would greatly underestimate the effects on slope stability. A proper estimation would require the simulation of flowpaths, by a more complex hydrological model, better representing the impact of heterogeneity, and is beyond the scope of this study. The manuscript here focuses only on rainfall infiltration creating pressure front and atmospheric effects on the pore pressure – which are non-hydrostatic processes. This hypothesis is stated during the model description and the seepage is mentioned in the model limitations.**

    ⇨ **A brief description of the seepage destabilising effect and why it is not computed in our model is provided in the section describing the water table model (lines 138-141).**

2D modeling of the homogeneous hill suggests that groundwater is shallower near the slope toe and deeper near the slope crest, as is well known. This initial condition strongly affects atmospheric- and rainfall-induced pore pressure change timing and magnitude along the height of the hillslope, as the manuscript demonstrates. The authors note that, for one typhoon, landslides in Taiwan concentrated near the lower parts of the slope, and they propose that this at least partly resulted from the shallower groundwater depth there. However, much of the preceding text noted that slope toes are more likely than upper parts of slopes to be saturated from long-term conditions, and if saturated, rainfall has no effect on stability and atmospheric pressure change will be of primary importance. Why would atmospheric pressure change in saturated regions not have been responsible for the landslide distribution? Additionally, such hillslope groundwater distribution should be ubiquitous, so does not the paper imply that rainfall/atmospheric-pressure-induced landslides everywhere should concentrate on lower parts of slopes? Can the authors provide evidence for this? Finally, 3 typhoons in Taiwan are mentioned. It would be beneficial if the authors described how pre-storm rainfall for the 3 events may have resulted in different landslide distributions, in accordance with their model.

5. **The idea behind these observations is indeed that the landslides distribution could be induced by atmospheric pressure changes; since infiltration – and therefore rainfall-induced pore pressure – is prevented in fully saturated areas. However, as already stated the model focuses on the dynamic pore pressure effects, leaving aside seepage or more complex effect, and disregarding the hydrostatic loading, which also may decrease stability in these areas.**
       ⇨ **We now more clearly state that the landslides distribution towards the toe of the hillslopes indicates an atmospheric-driven failure (lines 489-491).**

    **Taiwan has been chosen for the large number of typhoons, provoking many large atmospheric disturbances. The repartition of landslides towards the toe of the slope is indeed not limited to this area, but a general trend for landslides triggered by weather events. In case of earthquake-triggered landslides, slope failures tend to focus on the crest rather than the toe of the slope. These observations can be found in (Meunier et al., 2008) already cited in the manuscript.**
    **Among the 3 typhoons showed, only Morakot led to massive amounts of landslides. Masta and Krosa generated mudslides and debris flow, but in less quantity. No study has been conducted on the landslides and slope failure distribution along the hillslope after all these events.**
    **However, the model suggests such events over partially saturated slopes would have led to landslides near the toe of hillslopes during the typhoons, due to the atmospheric effect, then deeper landslides towards the upper unsaturated part of the hillslope after the typhoon due to the diffusion of rainfall-induced pore pressure. This has to be taken with caution, as the model does not represent all destabilizing processes – such as seepage – and should be verified with observed data.**

Please see the accompanying mark-up for additional comments and suggestions to improve the manuscript.

**The suggestions and comments have been taken into consideration in the manuscript.**

---

## Referee Report (RR1)

**Review of "Finite-hillslope analysis of landslides triggered by excess pore water pressure: the roles of atmospheric pressure and rainfall infiltration during typhoons"**

**Summary:**

I have read the response to reviewers and revised manuscript, and I am satisfied with the changes that have been made. While the model remains the same as that presented in the original manuscript, my primary conceptual concerns have been met with improvements to the text that describe the model behavior and express its limitations. The paper makes a compelling case that there are circumstances in which atmospheric pressure fluctuations could drive landslides during typhoons, and more generally that finite-hillslope analysis is a useful tool for understanding the mechanisms that generate landslides from excess pore pressure. I think it is a worthy and interesting contribution, however I will recommend a few editorial changes below.

**Line by line:**

**Title:**
I'm honored that you have used the title I suggested, thank you.

**Abstract:**
Line 13: "over" -> "on"

**Introduction:**
Line 31-32: What is "topographic site effect"?
Line 46: "Indeed" may be unnecessary. Check this and other locations as well.

**Method:**
Line 162: "strong pore pressure" might be clearer as "significant pore pressure response"

**Results – Synthetic:**
Line 260: "If" -> "While"

**Results – Application:**
Line 306: In many cases you have fully saturated hillslopes under diffusivities that seem within the reasonable range. It might be helpful to mention here or in discussion that reductions in recharge due to interception, evapotranspiration, or deep percolation could play important roles in the water balance and consequently could affect antecedent water table conditions (Herwitz 1985, Jasechko et al. 2014, Tromp-van Meerveld et al. 2007).

**Discussion:**
Between sections 5.4-5.7, I would suggest reviewing the organization and structure to be more concise and place related ideas together. For example, 5.7 supports the ideas presented in 5.5 so perhaps 5.5 could include 5.7 as evidence.

Lines 364-465: Check end of sentence: "… as for the finite hillslope model."
Lines 368-369: This would be a useful piece of information to have in the figure description too.
Line 491: "Phenomenon" -> "phenomena"
Figure 8: caption "repartition" -> "distribution"

Works Cited:

Herwitz, S. R. (1985). Interception storage capacities of tropical rainforest canopy trees. *Journal of Hydrology*, *77*(1), 237–252. https://doi.org/10.1016/0022-1694(85)90209-4

Jasechko, S., Birks, S. J., Gleeson, T., Wada, Y., Fawcett, P. J., Sharp, Z. D., et al. (2014). The pronounced seasonality of global groundwater recharge. *Water Resources Research*, *50*(11), 8845–8867. https://doi.org/10.1002/2014WR015809

Tromp-van Meerveld, H. J., Peters, N. E., & McDonnell, J. J. (2007). Effect of bedrock permeability on subsurface stormflow and the water balance of a trenched hillslope at the Panola Mountain Research Watershed, Georgia, USA. *Hydrological Processes*, *21*(6), 750–769. https://doi.org/10.1002/hyp.6265

---

## Referee Report (RR2)

[referee-annotated manuscript omitted]

---

## Author Response (AR2)

**Response to Referee 1**

I am recommending minor revisions focused on organization of discussion and acknowledgement of losses between precipitation and recharge.

**We are grateful for the review, and respond to the comment below.**

**The difference between rainfall and the recharge was not properly addressed. Indeed, not the whole volume of precipitations goes as water table recharge. Some is lost through runoff and some through evapotranspiration. The former effect was taken into account by limiting the infiltration rate to the hydraulic conductivity, and the latter was considered negligible at our timescale. This is now explicitly discussed (lines 125-130).**

**Response to Referee 3**

An analysis of landslides triggered by excess pore water pressure is described in the paper, with particular reference to typhoon events in Taiwan. The topic is certainly of interest to NHESS, and the work contains interesting data and considerations. I have listed in the accompanying file a number of small corrections, and a few requests of clarification on some issues that are not clear to me.

**We are grateful for the detailed review provided by Referee #3. We answer to the main comments below.**

In many parts of the manuscript (starting from the intro) Authors refer to catastrophic landslides. Nevertheless, they never define what a catastrophic landslide is. Is it a landslide of great volume and high velocity? Or is it a slope movement causing severe damage? Or what? In some sections, catastrophic landslides are in some ways counterposed to slow-moving landslides. Is it therefore just a matter of velocity? This should be clarified at the beginning of the article, by defining what a catastrophic landslide is.

**Catastrophic landslides are indeed opposed to slow-moving landslides. A catastrophic landslide is characterized by the suddenness of the failure and movement. By definition they are therefore more susceptible to cause damages and losses. A short clarification has been added in the introduction (lines 36-37)**

The introduction is well written and clear. However, I found lack of reference to an issue which I consider of extreme relevance to landslides, especially as concerns slope movements induced by rainfall: namely, weathering processes. These very often predispose the material to what is described in the introduction, that is changes in infiltration capacity and pore pressure. Therefore, a couple of sentences and some references should be added to this part in order to make it more complete. This is particularly important also given the study area (Taiwan) where the presence of weathered materials is particularly diffuse.

Thus, as concerns weathering, I suggest in particular to have a look at the chapters in the Engineering Geology Special Publication no. 23 by the Geological Society of London (2010, edited by Calcaterra and Parise) and at the references therein.

**This is a very good point, we agree that weathering plays a critical role in slope stability by changing the material mechanical and hydrological properties. However, the effect of weathering is a slow process that comes into play on long time scales. The aim of the study being the effect of typhoons on slope stability, we focus on short time scale phenomena. Rock properties during typhoons are considered constant and no spatial heterogeneity is introduced into the hillslope.**

**This being said, we agree the weathering process needs to be acknowledged – especially given the study area. It indeed could explain the observed landslides distribution towards the hillslope toe, where the shallower water table leads to stronger weathering. We mentioned the weathering process lines 33-34 & 83-87.**

Specific comments:

Line 121: there are quite generic definitions, you should provide some numbers, at least as an order of magnitude, in order to let the readers understand what you mean by gentle and steep. Further, it is not clear to me what you mean by narrow hillslopes. Please clarify this point.

**This is related to the Dupuit-Forchheimer hypothesis. Since the flow is considered horizontal only and transmissivity constant, the aquifer needs to be large and thick. This may not properly represent small hillslopes inside small catchments, where there is a lot of flow convergence. We now explain in more details this part of the hypothesis lines 131-137.**

Lines 183-185: talking about mechanisms of catastrophic landslides and slow-moving landslides, it is not very clear to what mechanisms are you referring to. This should be clarified, tpp.

**The mechanisms we referred to were the failure mechanisms, as the coulomb criterion and the safety factor are the same for slow-moving and catastrophic landslides. It has been clarified (line 196)**

Lines 392: typhoon Morakot caused more than 10,000 landslides, but nothing is said about these landslides. What typology do they have, are mostly shallow or not, what materials are involved, etc... Some information are needed about this event.

**Additional information about the observed landslides and their distribution has been provided in section 5.7 about the Morakot event (lines 559-561).**

In section 4 Taiwan is introduced but without a real description of the main physical characters of the area (basic description of its geology, presence of complex weathered profiles, typology of landslides, triggering factors, etc.). This must be included in the manuscript.

**This was indeed necessary, a short paragraph has been added (lines 282-285).**

A brief text explaining the importance of establishing relationships between rainfall and geological hazards could be useful (see for instance at this regard the works by Peruccacci et al. 2012, Rossi et al. 2012, and Vessia et al. 2014).

**A sentence was added in the introduction to emphasize the importance of a better understanding of the role of weather events in risk assessments (lines 49-50).**

Suggested references:

Calcaterra D. & Parise M., 2005, Landslide types and their relationships with weathering in a Calabrian basin, southern Italy. Bulletin of the Engineering Geology and the Environment, vol. 64, no. 2, p. 193-207.

Calcaterra D. & Parise M., 2010, Weathering as a predisposing factor to slope movements: an introduction. In: Calcaterra D. & Parise M. (Eds.), Weathering as a predisposing factor to slope movements. Geological Society of London, Engineering Geology Special Publication no. 23, p. 1-4.

Peruccacci, S., Brunetti, M. T., Luciani, S., Vennari, C., and Guzzetti, F.: Lithological and seasonal control on rainfall thresholds for the possible initiation of landslides in central Italy, Geomorphology, 139–140, 79–90, 2012.

Rossi, M., Peruccacci, S., Brunetti, M. T., Marchesini, I., Luciani, S., Ardizzone, F., Balducci, V., Bianchi, C., Cardinali, M., Fiorucci, F., Mondini, A. C., Reichenbach, P., Salvati, P., Santangelo, M., Bartolini, D., Gariano, S. L., Palladino, M., Vessia, G., Viero, A., Antronico, L., Borselli, L., Deganutti, A. M., Iovine, G., Luino, F., Parise, M., Polemio, M., and Guzzetti, F.: SANF: a national warning system for rainfall-induced landslides in Italy, in: Proceedings of the 11th International Conference and 2nd North American symposium on landslides, Banff, Alberta, Canada, 3–8 June, 2012.

Vessia G., Parise M., Brunetti M.T., Peruccacci S., Rossi M., Vennari C. & Guzzetti F., 2014, Automated reconstruction of rainfall events responsible for shallow landslides. Natural Hazards and Earth System Sciences, vol. 14, p. 2399-2408.

---

## Author Response (AR3)

**Response to Editor**

The paper was reviewed and the comments were taken into account. However, introducing new text, as requested by the referees, was not properly accompanied by citations to support several of these statements. Some references have been suggested by referees, you could choose those (if pertinent) or others, but the new statements about links between rainfall events and landslides, or weathering and landslides, should be supported by some references to the available international literature

**We are grateful for the swift response of the Editor. Small modifications has been made to the manuscript and references regarding weathering, rainfall-induced landslides and the triggering mechanism of slow and catastrophic landslides have been added.**